# An Empirical Study on Enhancing LLMs' Alignment Capabilities through Restyled In-Context Learning Demonstration Examples

## Abstract

Alignment tuning is crucial for ensuring large language models (LLMs) behave ethically and helpfully. Current alignment approaches, including supervised fine-tuning (SFT) and preference optimization (PO), require high-quality annotations and significant training resources. This paper proposes a *low-cost, tuning-free method using in-context learning (ICL) to enhance LLM alignment*.

By comparing token generation probabilities of unaligned models with those of their aligned counterparts, we identified *polarity tokens*—a type of special token that guides the response trajectory and affects the model's alignment performance. Based on this, we designed heuristic rules to select ICL demonstration examples that effectively influence polarity token distributions. To further improve the quality of the selected ICL demonstrations, we model *content*, *style*, and *alignment* as a *causal structure*, and employ Average Treatment Effect (ATE) to quantitatively study the impact of the style factor. We found that after restyling the ICL demonstrations, they were more effective at eliciting the LLM's alignment capabilities.

We packaged the restyled examples as prompts to trigger few-shot learning, improving LLM alignment. Our experiments show that rewritten examples boost alignment, safety, and reasoning across various tasks. Compared to the best baseline approach, with an average score of 5.00 as the maximum, our method achieves a maximum 0.10 increase on the `Alpaca-eval` task (from $4.50 \rightarrow 4.60$), a 0.19 enhancement on the `just-eval-instruct` benchmark (from $4.44 \rightarrow 4.63$), and a maximum improvement of 0.32 (from $3.53 \rightarrow 3.85$) on the `MT-Bench` dataset. These findings underscore the need for deeper analysis and theoretical understanding of alignment for advancing future LLM research.

## 1 Introduction

Alignment tuning helps bridge the gap between raw model capabilities and the nuanced requirements of different tasks, such as delivering accurate information, maintaining user safety, and handling sensitive topics with care (Shneiderman, 2020; Shen et al., 2023; Wang et al., 2023; Qi et al., 2024b). Currently, the instruction-following paradigm (Ouyang et al., 2022; Sun et al., 2023; Dai et al., 2024; Rafailov et al., 2024; Zhou et al., 2024; Wu et al., 2024), which combines supervised fine-tuning (SFT) and preference optimization, is widely used in alignment tuning. However, this paradigm requires high-quality annotated data and consumes significant computing resources. If there were a method to improve LLM alignment without modifying model parameters, it could effectively reduce training costs and increase the versatility of the alignment process (Brown, 2020). To this end, we propose an in-context learning (ICL) method, which instructs LLMs to handle downstream tasks using few-shot learning from ICL demonstrations.

Many ICL demonstration selection methods have been proposed to choose demonstrations that effectively elicit the desired model capabilities (Liu et al., 2022; Min et al., 2022; Ye et al., 2023; Luo et al., 2023; Peng et al., 2024; Choi & Li, 2024; Wang et al., 2024), however, these methods are not suitable for alignment tuning due to its unique nature.

Alignment tuning imposes two conflicting demands on LLMs: on one hand, LLMs need to provide more in-depth, informative, and helpful content (*factuality*) (Shen et al., 2023); on the other hand, for

safety reasons, LLMs must refuse to answer inappropriate queries (*safety*) (Ji et al., 2024). Thus, we need to achieve a delicate **trade-off** between these two abilities (Anwar et al., 2024), as improving the harmlessness of an LLM assistant may result in it being less helpful (Bai et al., 2022). This trade-off presents a challenge when selecting demonstrations – the demonstrations must balance both factuality and safety. To address this, we investigate those special tokens that influence the generation trajectory and denote them as *polarity* tokens (Section 2). Within *polarity* tokens, **benign** tokens steer content generation towards helpful and constructive outputs, while **malicious** tokens can lead the trajectory towards harmful or undesirable outputs. Additionally, we find that the polarity tokens unique to *factuality* and *safety* are entirely different, which further validates that alignment imposes two distinct requirements on LLMs that need to be balanced.

Identifying the *polarity* tokens related to alignment tuning results in a powerful tool. By observing changes in the generation probability of polarity tokens, we can approximate whether an ICL demonstration effectively elicits LLM alignment capabilities (Section 3.1). However, we also need to consider how we can further enhance the alignment capability brought by an effective ICL demonstration that alters polarity token generation probabilities.

***Content*** (referring to the contextual information contained in a demonstration, including the system instruction and one-shot learning example) and ***style*** (referring to the format in which the content is organized and written with a specific style and structure) can impact in-context alignment (Huang et al., 2024b). We hypothesize that the *content* and *style* of an ICL example have a ***causal relationship*** with the alignment effectiveness. We consider *restyling* an ICL demonstration example as an *intervention* and use Average Treatment Effect (ATE) to determine the effect of different restyling methods on alignment. Using ATE, we design a method to learn how to restyle an ICL demonstration example to enhance its effectiveness (Section 3.2). We then combine these restyled ICL demonstrations to achieve a trade-off between factuality and safety (Section 3.3).

Evaluating our approach across multiple datasets and LLMs (Section 4) reveals that different benchmarks have distinct preferences regarding LLM alignment capabilities, and that the combinations of restyled ICL demonstrations can meet these diverse preferences. In summary, our contributions are three-fold:

- We propose polarity tokens, a class of tokens that guides the content generation trajectory of LLMs and influences their alignment performance. Specifically, under different alignment requirements, factuality and safety capabilities correspond to distinct polarity tokens. We use changes in the generation probability of polarity tokens as an indicator to retrieve high-quality ICL demonstration examples that can elicit LLM alignment capabilities.

- We model the content, style, and alignment capabilities of ICL examples as a causal structure and employ ATE method to quantitatively explore the effect of style on alignment, thereby identifying a way to further enhance the quality of ICL demonstrations. We combine the restyled demonstrations to balance factuality and safety, improving the LLMs' overall alignment performance.

- We conduct a series of experiments across different datasets and LLM models, demonstrating the effectiveness and superiority of our proposed method. The experimental results show that, across the three benchmarks, our method achieves improvements of $2.22\%$, $4.28\%$, and $9.07\%$ compared to the SOTA methods, respectively.

## 2 BACKGROUND

In this paper, the terms "unaligned LLMs" and "base LLMs" are interchangeably used to refer to LLMs that have not undergone preference optimization. In contrast, we refer to LLMs that have been fine-tuned with instructional data to promote ethical and beneficial behavior as "aligned LLMs". More details can be found in Appendix A.2.

### 2.1 THE CHANGE IN TOKEN GENERATION PROBABILITIES DUE TO ALIGNMENT TUNING

Based on the characteristics of autoregressive models, we adopt a unique perspective—examining the probability distribution of token generation—to study how alignment tuning affects and intervenes in the content generated by the model.

Compared to base LLMs, aligned LLMs, after alignment tuning, increase the generation probability of specific tokens when faced with the same queries—these tokens often guide the LLMs toward producing ethical and regulated responses, where we demoted as **benign tokens**. Conversely, the generation probabilities of other tokens, which are more likely to cause the LLMs to deviate toward content lacking alignment safeguards, are reduced. We call these tokens as **malicious tokens**.

To understand how the "***alignment tuning***" process affects token generation and to identify which tokens can be classified as **benign** or **malicious**, we propose analyzing the discrepancy between the token generation probabilities of base LLMs and aligned LLMs. Specifically, for a given user query $\mathbf{q} = \{q_1, q_2, \cdots\}$, we input it into an reference model $r(x)$ to obtain its output $\mathbf{o} = \{o_1, o_2, \cdots\}$ via greedy decoding. For each position $t$, we define a *'context'* at this position to be $\mathbf{x_t} = \mathbf{q} + \{o_1, \cdots, o_{t-1}\}$. We denote the reference model's probability distribution for predicting the next token of this position as $P_t^{\texttt{refer}}$, where $o_t$ has the highest probability. Our analysis is driven by the following questions: *If we take the base model $f$ as the reference model and compare the output of the aligned model $g$ to it, what kind of differences in probability distribution would we observe? Conversely, if we treat the aligned model $g$ as the reference model and compare the unaligned model $f$ to $g$, what distinctions can we see?*

We first view the unaligned model $f$ as the reference model, and pass the context $\mathbf{x_t}$ into the base model $f$, we generate the reference output $\mathbf{o}_t$ and the corresponding output probability distribution $P_t^{\texttt{base}}$ for sampling the next token at this position. Let $\mathbf{o}_t$ represent the context for the $t$-th position, thus we compute the probability distribution $P_t^{\texttt{align}}$ of the aligned model $g$ at the this position. If alignment tuning enables the LLM to learn how to adjust the probability distribution over the output vocabulary to prevent the model from favoring tokens that may lead to malicious content, then by comparing $P^{\texttt{base}}$ and $P^{\texttt{align}}$, we should observe a noticeable decrease in the generation probability of certain tokens.

Similarly, if we treat the aligned model $g$ as the reference model and use its output as the reference $\mathbf{o}$, we can force the unaligned model $f$ to generate $\mathbf{o}$, and during this process, observe the shifts in probability distributions between $P^{\texttt{align}}$ and $P^{\texttt{base}}$.

**Polarity tokens.** When we consider the unaligned model $f$ as the reference model, its generated content tends to lack alignment safeguards and may be more harmful. As a result, in the earlier positions of such output (Qi et al., 2024a), malicious tokens dominate, steering the generation towards harmful paths. In contrast, the aligned model $g$, after alignment tuning, should lower the generation probabilities of these malicious tokens, pulling the output back onto the trajectory that is more in line with the human values.

Therefore, when using the output $\mathbf{o}_t$ from the unaligned model $f$ as the reference, we can calculate the probability distribution differences across the output vocabulary at position $t$. This is expressed as: $\Delta P_t^{\texttt{malicious}} = P_t^{\texttt{base}} - P_t^{\texttt{align}}$. For a validation dataset containing multiple $\mathbf{q} - \mathbf{o}$ pairs, we count all the tokens present in the reference outputs and calculate the average $\Delta P_t^{\texttt{malicious}}$ for each unique output token $\mathbf{o}_t$. The output tokens with the highest average $\Delta P_t^{\texttt{malicious}}$ value are considered as **malicious tokens**.

Likewise, we view the aligned model $g$ as the reference, and its output $\mathbf{o}$ as the ground-truth output. We thus compute $\Delta P^{\texttt{benign}} = P^{\texttt{align}} - P^{\texttt{base}}$, and consider this difference as the adjustment to the generation probabilities made through alignment tuning. After tuning, the aligned model $g$ tends to favor tokens that guide the generated content toward helpful and harmless outputs, whereas the probability of these tokens is relatively low in the output of the unaligned model $f$. We also viewed the tokens with the highest average difference $\Delta P^{\texttt{benign}}$ as **benign tokens**.

## 2.2 FINDINGS OF POLARITY TOKENS

The alignment process for LLMs ensures that AI models are not only technically proficient but also socially responsible, making them more suitable for real-world applications where trust and reliability are critical. Therefore, LLMs that have undergone alignment tuning must generate factually accurate information across various tasks (referred to here as "*factuality*") while also refusing to respond to malicious queries (referred to here as "*safety*") (Shen et al., 2023). However, these two capabilities present a "***conflict***" in terms of LLMs' token generation preferences (Tuan et al., 2024). Specifically, when providing useful information, LLMs tend to generate tokens that convey a pos-

itive attitude in the initial stages of autoregressive generation (e.g., 'Let's', 'Here's'). Conversely, when refusing to respond, the probability distribution of generated tokens shifts toward polite and apologetic expressions (e.g., 'sorry', 'condone', 'cannot'). Analyzing these two capabilities together reduces the distinctiveness of the distributions, thereby increasing the difficulty of analysis.

To address this "*conflict*", in this work, we designed two sub-tasks to facilitate a clearer analysis of polarity tokens associated with LLMs' different alignment capabilities. The first sub-task focuses on analyzing the polarity tokens that enable LLMs to generate factually accurate information. The second sub-task involves analyzing the polarity tokens that enhance the LLM's safety capabilities.

In our empirical study of polarity tokens, we selected `llama-2-7b` as the base LLM and used `llama-2-7b-chat` as its aligned counterpart. We utilized the `Alpaca-eval` benchmark (Dubois et al., 2024) to analyze polarity tokens related to LLM helpfulness, while a safety-related subset of the `just-eval-instruct` dataset (Lin et al., 2024) was used to analyze safety polarity tokens.

As described in Section 2.1, for the two sub-tasks, we separately calculated $\Delta P_{\texttt{malicious}}$ and $\Delta P_{\texttt{benign}}$, thereby identifying the polarity tokens that influence factuality and safety.

| Task | Benign | Malicious |
|------|--------|-----------|
| *Factuality* | is, a, and, that, The, can, asking, As, in, are, by, for, of, Thank, me | to, I'm, you, Let's, the, this, I, help, closer, look, understand, Hello!, unable, sorry, Here's |
| *Safety* | I, but, am, sorry, cannot, condone, or, provide, you, assist, Dear, in, how, I'm, understand | a, is, are, type, cyberattack, can, requires, attacks, where, be, never, careful, there!, Hi, Making |

Table 1: The top-15 polarity tokens for two sub-tasks.

In Table 1, we present the top-15 typical polarity tokens in descending order based on the magnitude of $\Delta P$. From the table, we can see that the factuality and safety sub-tasks have their own preferred polarity tokens. Based on this finding, we utilized two distinct sets of polarity tokens for factuality and safety, respectively, as the foundation for the ICL selection described in Section 3.1.

Further analysis of polarity tokens, as well as discussions on the relationship between polarity tokens and alignment sub-tasks (factuality and safety), can be found in the Appendix A.3.

## 3 ICL DEMONSTRATIONS FOR LLM ALIGNMENT

The analysis in Sec. 2.1 inspired us to make the following hypothesis: if an In-context Learning (ICL) example positively influences the probability distribution of polarity tokens, then it should be considered a high-quality ICL demonstration example. The following three questions then arise: *how can we use an objective, automated metric to identify a single high-quality demonstration example (Section 3.1)? If we can further enhance the quality of this example, will it make it even more beneficial for ICL (Section 3.2)?* Multiple ICL demonstrations are often more effective than a single ICL example; *how can we identify such an approximate optimal demonstration set (Section 3.3)?*

### 3.1 SELECTION OF SINGLE ICL DEMONSTRATIONS

We propose the following hypothesis: a "good" ICL demonstration example should enable the model to significantly adjust the generation probabilities of polarity tokens, increasing the probability of generating benign tokens while reducing the probability of generating malicious tokens.

For a given user query $\mathbf{q} = \{q_1, q_2, \cdots\}$, to calculate the impact of an ICL example $c$ on benign tokens, we use the aligned model $g$ to generate a reference output $\mathbf{o}$. For all benign tokens $\{\mathbf{o}_b\}$ in $\mathbf{o}$, we compute the enhancement in the generation probability of benign tokens by an ICL example using the following formula: $\Delta P_{\mathtt{b}}(q, c) = \Sigma_{o_t \in \{\mathbf{o}_b\}} \alpha_t [P^{\mathsf{base}}(o_t|x_t, c) - P^{\mathsf{base}}(o_t|x_t)]$.

Here, $\alpha_t$ is used to control the importance weight of the probability difference. We believe that the initial few tokens generated by the model have a greater impact on the trajectory of the generation. Therefore, we assign higher weights $\alpha_t$ to the tokens $o_t$ at the early positions, and gradually decrease the weight $\alpha_t$ as the generation progresses.

Similarly, to calculate how an ICL example $c$ reduces the generation probability of **malicious tokens**, we first use the unaligned model $f$ as the reference to generate the output $\mathbf{o}$. We then compute the effect on each malicious tokens $\{\mathbf{o}_m\}$ in $\mathbf{o}$ individually, yielding: $\Delta P_{\mathtt{m}}(q,c) = \Sigma_{o_t \in \{\mathbf{o}_m\}} \alpha_t [P^{\mathsf{base}}(o_t|x_t) - P^{\mathsf{base}}(o_t|x_t,c)]$.

If an ICL example can have the higher value: $V_{polar} = \frac{\Sigma_{q_t \in Q}(\Delta P_{\mathtt{b}}(q_t,c) + \Delta P_{\mathtt{m}}(q_t,c))}{|Q|}$ for the examples in the validation dataset $Q$, it should be an ICL example that effectively adjusts the generation probabilities of polarity tokens, thereby guiding the LLM's generation trajectory more towards alignment with human values. We choose the ICL demonstration examples that have the highest $V_{polar}$ as the high-quality ICL examples.

In empirical study, we treated the QA pairs from UltraChat (a large-scale multi-turn dialogue corpus aimed at training and evaluating advanced conversational AI models) (Ding et al., 2023) and SORRY-Bench (a dataset intended to be used for LLM safety refusal evaluation) (Xie et al., 2024) as candidate pools for ICL demonstration examples of factuality and safety, respectively.

From Section 2.2, we knew that polarity tokens for factuality and safety are different. Therefore, we used these two distinct sets of polarity tokens to select the top-20 ICL demonstration examples with the highest $V_{polar}$ for the factuality and safety subtasks, respectively, as $\{S_{\mathrm{cand\_f}}\}$ and $\{S_{\mathrm{cand\_s}}\}$.

## 3.2 A Causal Approach to Restyle

In this section, we hypothesize that restyling can further enhance the quality of ICL demonstrations, and we validate this hypothesis through experiments.

We first provide the following definitions. ***content*** refers to the task-related information provided in an ICL example, including the system instruction and the demonstration, ***style*** represents the writing style of task-related information and the organizational structure of the content, and ***alignment*** refers to the alignment effect exhibited by the model after using a particular example as a ICL demonstration.

We consider ***style*** and ***content*** to be the two most critical factors in applying ICL techniques for alignment tuning. We model $S$ (*style*), $C$ (*content*), and $A$ (*alignment*) as a ***causal structure*** (Pearl, 2009), as illustrated in Figure 1. The variable $C$ is the co-founder, which influences both $S$ and $A$. Both $C$ and $S$ jointly influence *alignment*.

**Content**  We consider $C$ as a factor that cannot be experimentally manipulated. On the one hand, using LLMs to modify the content of an LLM's response can lead to hallucinations, making the study uncontrollable. On the other hand, altering the content changes the nature of the demonstration, thus losing the significance of the research. Therefore, our primary interest lies in the impact of the intervenable factor $S$ on $A$, and we thus disregard the influence of $C$ on $A$, focusing instead on evaluating the effect of the controllable intervention $S$.

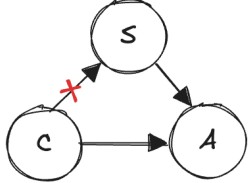

Figure 1: The causal structure of style, content, and alignment.

**Style**  To quantify the impact of an intervention on an outcome of interest, the Average Treatment Effect (ATE) is a commonly used method in causal inference (Kaddour et al., 2021; Mahajan et al., 2024). Therefore, we use ATE as the expected difference in outcomes to determine, on average, how much effect the intervention has compared to other interventions.

Specifically, following the principles of causality, we consider setting $S$ to a fixed value as an intervention, denoted using the *do*-operator: $do(S = s)$[1]. Whenever $do(s)$ appears after the conditioning bar, it means that everything in that expression is in the *post-intervention* world where the intervention $do(s)$ occurs.

It is important to note that, in Figure 1, there is an edge from $C$ to $S$, indicating that $C$ confounds the effect of $S$ on $A$. However, according to the definition in causal theory, $do(s)$ will remove the

---

[1]Which can also be shortened to $do(s)$.

edge from $C$ to $S$ when intervening on $S$, meaning that $C$ will no longer affect $S$, as indicated by the red cross in the figure.

Thus, $E(A|do(S = s))$ refers to the expected alignment improvement after all examples have been restyled using the format $s$. According to the backdoor criterion, we obtain:

$$E[A|do(S = s)] = \sum_c E[A|s, C = c]p(c) \tag{1}$$

The ATE is defined as:

$$ATE(s_t, s_o) = E[A|do(S = s_t)] - E[A|do(S = s_o)] \tag{2}$$

where $s_t$ refers to target style, and $s_o$ denotes other style.

Empirically, we adopted the idea of Monte Carlo sampling (Knaus et al., 2021) and approximate $p(c)$ as a uniform distribution. We used a single example as the ICL demonstration, enabling the LLM to handle downstream tasks through one-shot online learning. To calculate the expectation $E[A|s, C = c]$, we kept the content of the ICL demonstration fixed ($C = c$), while restyling the demonstration example with a specific style $s$. The restyled demonstration example is then encapsulated in the prompt and fed to the LLM, which processes examples from the validation dataset via ICL. We considered the LLM's average alignment performance on the validation dataset as an approximation of $E[A|s, C = c]$.

Based on the concept of Monte Carlo sampling, we randomly selected $N$ ICL demonstrations[2] from the candidate high-quality ICL examples to form the set $\{C\}$. Corresponding to the $N$ demonstrations in $\{C\}$, we applied the same restyle process to each, resulting in $N$ average alignment performance values. By averaging these $N$ values, we obtain an approximation of $E[A \mid do(S = s)]$, where $c \in \{C\}$ and $p(c)$ follows a uniform distribution. It is worthy noting that in Section 2.2, we found LLMs exhibit conflicting behavior when handling "*factuality*" and "*safety*" sub-tasks. In such cases, the LLM needs to achieve a trade-off between these two capabilities to mitigate the conflict. Therefore, we randomly selected a set $\{C_f\}$ from $\{S_{cand.f}\}$ (defined in Section 3.1), focusing on "*factuality*", and a set $\{C_s\}$ from $\{S_{cand.s}\}$ (defined in Section 3.1), focusing on "*safety*", and applied the same style restyling to each.

In our work, we used a powerful LLM[3] to modify the style of the answer part in the following ways: (1) three-part (presenting the answer in a three-part structure: first, introducing the answer in one sentence; second, itemizing the answer using bullet points; and third, summarizing the answer in one sentence), (2) lengthy (enriching the answer details and increasing its length without altering the original meaning), (3) human (using a conversational tone or answering from a first-person perspective), (4) combined (use three-part, lengthy and human three styles to rewrite the ICL example simultaneously), (5) refusal (for safety-related ICL examples, first refuse to answer, then provide a reason, and finally offer advice or guidelines), and (6) no style (the original ICL demonstration that remains unchanged).

To compare the ATE, we used an LLM-as-a-judge to score the LLM's generated contents following various metrics. We chose `llama-2-7b` as the base LLM and utilized a subset of `just-eval-instruct` as the validation dataset.

By analyzing the ATE results (details can be found in Appendix A.4), we have the following findings: (1) for factuality-related ICL demonstration examples, we should adopt the "combined" style; (2) for safety demonstrations, the "refusal" style should be used; (3) the differing emphases of the factuality and safety subtasks on various styles validate our findings in Section 2.2, namely, that to achieve optimal overall performance in an LLM, a trade-off between factuality and safety must be reached. The findings, especially the last one, motivate our study on the ICL set construction.

## 3.3 Selection of a Set of ICL Demonstrations

Finding an optimal demonstration example set from the candidate high-quality ICL examples is a NP-hard problem (Ye et al., 2023), and so heuristic approaches should be used in general to get an (approximate) optimal approximation solution (Liu et al., 2024).

---

[2]To reduce computational complexity, we set $N$ to 5.

[3]We used GPT-4o to restyle the answers in the ICL examples.

Previous research has shown that subtle interactions between the demonstrations in an ICL example set can significantly influence the performance of LLMs in few-shot online learning. On the one hand, maintaining a consistent response style across ICL demonstration examples can effectively enhance LLM performance on downstream tasks (Lin et al., 2024; Li et al., 2024). On the other hand, the multiple ICL demonstrations needs to be sufficiently diverse and complementary to fully elicit LLMs' task-oriented capabilities (Min et al., 2022). Notably, when dealing with safety tasks, having refusal demonstration in the set becomes particularly crucial.

In Section 3.1, we already formed candidate sets $\{S_{\text{cand\_f}}\}$ and $\{S_{\text{cand\_s}}\}$. Therefore, for the *factuality* candidates $\{S_{\text{cand\_f}}\}$, we restyled them using the "combined" style, while for the *safety* candidates $\{S_{\text{cand\_s}}\}$, we restyled them using both the "combined" and "refusal" styles. To achieve the optimal trade-off between factuality and safety, we merged the restyled *factuality* and *safety* candidates into a set $\{S_{\text{cand}}\}$ and employed a hierarchical traversal approach with early pruning (Hua et al., 2024) to select three ICL examples[4] from $\{S_{\text{cand}}\}$ to construct different demonstration sets. We evaluated the performance of different combinations on the `just-eval-instruct` validation dataset, like what has been done in Section 3.2.

Ultimately, we identified the three best combinations of the ICL examples. The first combination consists of three factuality ICL examples restyled with the "combined" style. The second combination includes two factuality ICL examples and one safety example, all restyled using the "combined" style. The third combination consists of two factuality ICL examples restyled with the "combined" style and one safety example restyled with the "refusal" style. We refer to these combinations as Restyled In-context-learning Demonstration Examples (RIDE), with the first combination denoted as $\mathbf{RIDE}_{\text{f}}$, the second as $\mathbf{RIDE}_{\text{fs\_uni}}$, and the third as $\mathbf{RIDE}_{\text{fs\_hyb}}$. We use these notations in the following sections. The prompts of $\mathbf{RIDE}$ series can be found in Appendix A.6, A.7, and A.8.

## 4 EVALUATION

### 4.1 DATASET, LLMs, AND BASELINE METHODS

**Dataset.** We use `Alpaca-eval` (a benchmark designed to assess the performance of language models on natural language understanding, generation, and reasoning tasks) (Li et al., 2023), `just-eval-instruct` (a dataset designed to assess the ethical reasoning capabilities of LLMs) (Lin et al., 2024), and `MT-Bench` (a multi-turn dialogue dataset to evaluate various capabilities of LLMs, such as reasoning, coding, and human-like interaction) (Zheng et al., 2023) as benchmarks.

**LLMs.** We use three models as the base models: Llama-2-7b-hf (Touvron et al., 2023), Mistral-7b-v0.1 (Jiang et al., 2023), and OLMo-7B (Groeneveld et al., 2024). It is important to note that these models have not undergone alignment tuning, resulting in sub-optimal alignment capabilities.

**Baseline methods.** We selected different baseline methods for comparison. The most relevant to our work is URIAL (Lin et al., 2024), which manually designed three ICL examples and combined them with system instructions (explicitly informing the LLM to generate safe and reliable content) to form prompts, achieving state-of-the-art (SOTA) performance across multiple datasets using the ICL approach. Additionally, we compared against the following baselines: (1) **Zero-shot**: consisting only of the URIAL system instruction part. (2) **Vanilla ICL**: an ICL example set composed of the top-2 examples from $\{S_{\text{cand\_f}}\}$ and the top-1 example from $\{S_{\text{cand\_s}}\}$. (3) **Retrieval ICL** (Liu et al., 2022): Among the examples in $\{S_{\text{cand}}\}$, the neighbors that are the most similar to the given test query are retrieved as the corresponding in-context examples. (4) **TopK + ConE** (Peng et al., 2024): a tuning-free method that retrieves the best three examples that excel in reducing the conditional entropy of the test input as the ICL demonstrations. In this work, we consistently use GPT-4o as the LLM-as-a-judge to evaluate and score the responses generated by the LLMs. Through comparing these baseline methods with our proposed ICL demonstration set, i.e., $\mathbf{RIDE}_{\text{f}}$, $\mathbf{RIDE}_{\text{fs\_uni}}$, and $\mathbf{RIDE}_{\text{fs\_hyb}}$, we conducted a detailed experimental analysis.

---

[4]To reduce the search space while maintaining a sufficient number of ICL demonstrations, and to align with the number of ICL examples used in SOTA URIAL method (ensuring a more straightforward comparison in experiments), we set the number of ICL demonstrations to 3.

## 4.2 Empirical results on `just-eval-instruct`

| Models + ICL Methods | Helpful | Factual | Deep | Engaging | Clear | Safe | Average | Length |
|---|---|---|---|---|---|---|---|---|
| Llama2-7b + **Zero-shot** | 2.94 | 2.79 | 2.57 | 3.66 | 3.65 | 2.24 | 2.98 | 211.99 |
| Llama2-7b + **Vanilla ICL** | 3.21 | 3.26 | 2.85 | 4.00 | 3.96 | 2.55 | 3.31 | 224.52 |
| Llama2-7b + **Retrieval ICL** | 3.27 | 3.19 | 3.17 | 4.04 | 3.87 | 2.75 | 3.38 | 229.17 |
| Llama2-7b + **TopK + ConE** | 3.44 | 3.45 | 3.20 | 4.02 | 4.16 | 2.80 | 3.51 | 226.11 |
| Llama2-7b + Urial | 3.98 | **3.98** | 3.64 | 4.36 | 4.52 | 4.42 | 4.15 | 239.81 |
| Llama2-7b + **RIDE**$_f$ | **4.09** | 3.87 | **3.82** | **4.52** | **4.56** | 2.81 | 3.95 | **303.41** |
| Llama2-7b + **RIDE**$_{fs\_uni}$ | 3.90 | 3.90 | 3.64 | 4.34 | 4.48 | 4.17 | 4.07 | 266.76 |
| Llama2-7b + **RIDE**$_{fs\_hyb}$ | 3.95 | 3.95 | 3.69 | 4.40 | 4.52 | **4.45** | **4.16** | 238.05 |
| Mistral-7b + Urial | 4.41 | 4.43 | 3.90 | 4.57 | 4.79 | **4.89** | 4.50 | 214.60 |
| Mistral-7b + **RIDE**$_f$ | **4.67** | **4.49** | **4.42** | **4.75** | **4.85** | 4.13 | 4.55 | **304.51** |
| Mistral-7b + **RIDE**$_{fs\_uni}$ | 4.59 | 4.44 | 4.27 | 4.69 | 4.83 | 4.50 | 4.55 | 289.19 |
| Mistral-7b + **RIDE**$_{fs\_hyb}$ | 4.58 | 4.43 | 4.16 | 4.63 | 4.83 | **4.89** | **4.60** | 252.69 |
| Olmo-7b + Urial | 3.45 | 3.62 | 3.13 | 3.94 | 4.20 | **2.70** | **3.51** | 203.86 |
| Olmo-7b + **RIDE**$_f$ | **3.52** | 3.57 | **3.20** | **4.10** | **4.27** | 1.79 | 3.41 | **225.31** |
| Olmo-7b + **RIDE**$_{fs\_uni}$ | 3.46 | 3.61 | 3.14 | 3.93 | 4.25 | 2.44 | 3.47 | 200.92 |
| Olmo-7b + **RIDE**$_{fs\_hyb}$ | 3.44 | **3.65** | 3.08 | 3.88 | 4.20 | 2.69 | 3.48 | 189.96 |

Table 2: **Multi-aspect scoring evaluation of alignment methods on `just-eval-instruct`.** Each block is corresponding to one specific LLM. Scores are on a scale of 1-5. **Average** refers to the averaged score of the 6 metrics and **Length** is computed by number of words.

Table 2 presents the scores of each method on `just-eval-instruct`. In `just-eval-instruct`, the dataset places a great emphasis on safety. Out of the 1000 test cases in `just-eval-instruct`, 200 questions are safety-related and require the model to provide clear refusal responses. The remaining 800 instances are related to factuality, requiring the LLM to provide accurate and helpful factual knowledge. Therefore, `just-eval-instruct` evaluates both the factuality and safety capabilities of the LLM, requiring the LLM to make a balanced trade-off between the two.

From Table 2, we can summarize the following conclusions. First, among the three proposed ICL sets, **RIDE**$_{fs\_hyb}$ performs the best, followed by **RIDE**$_{fs\_uni}$, and finally **RIDE**$_f$. **RIDE**$_{fs\_hyb}$ includes both factuality and safety ICL examples, with the safety demonstration restyled using the "refusal" style, which effectively enhances the LLM's safety capability while maintaining good factuality. Although **RIDE**$_{fs\_uni}$ also contains a safety demonstration, it uses the "combined" style for restyling. While the three examples in it have a consistent style, the safety ability of the safety example is weakened, resulting in a lower "Safe" score compared to **RIDE**$_{fs\_hyb}$. As for **RIDE**$_f$, which consists entirely of factuality examples, it has the strongest factuality capability but lacks any safety example, preventing the LLM from learning how to refuse malicious queries, leading to a much lower "Safe" score compared to the other two ICL sets. This finding aligns with our observations in Section 3.2.

Second, compared to Urial, **RIDE**$_{fs\_hyb}$ outperforms it in two out of three models. In the case of OLMo-7B, the input window length is severely limited (only 2048 tokens), while our prompts containing ICL examples exceed this limit. Thus, we had to randomly remove parts of the ICL bullet points, which especially affects the LLM's performance in "Helpful", "Factual", and "Deep". However, even under such constraints, we can see that **RIDE**$_{fs\_hyb}$ performs comparably with Urial in various aspects, with nearly identical scores in the crucial "Safe" metric (2.69 vs 2.70), although it is slightly weaker in the overall "Average" score (3.48 vs 3.51).

Third, in the first block of Llama2-7b, we compared four baseline methods. It can be observed that the baseline methods exhibit a significant performance gap compared to Urial and our ICL sets. **TopK + ConE** is the closest in principle to our approach: selecting good ICL demonstrations by observing the impact of ICL on content generation during inference. **TopK + ConE** is the best among the four baseline methods, but there is still a considerable gap compared to our approach.

In addition, comparing **Vanilla ICL** and our **RIDE** series ICL example sets reveals that simply combining the best-performing examples from $\{S_{cand\_f}\}$ and $\{S_{cand\_s}\}$ does not yield an optimal set. The performance gap between **Vanilla ICL** and **RIDE** also validates the effectiveness of our method for selecting a set of ICL demonstrations, as described in Section 3.3. Noted that we only used

Llama-2-7b-hf in this benchmark to compare all baseline methods and evaluate their performance to reduce token consumption when calling LLM-as-a-judge.

## 4.3 EMPIRICAL RESULTS ON Alpaca-eval

| Models + ICL Methods | Helpful | Factual | Deep | Engaging | Clear | Safe | Average | Length |
|---|---|---|---|---|---|---|---|---|
| Llama2-7b + URIAL | 3.82 | 3.88 | 3.52 | 4.26 | 4.45 | **4.89** | 4.14 | 238.67 |
| Llama2-7b + **RIDE$_f$** | **3.98** | 3.84 | **3.68** | **4.39** | **4.49** | 4.87 | **4.21** | **283.62** |
| Llama2-7b + **RIDE$_{fs\_uni}$** | 3.87 | 3.89 | 3.55 | 4.26 | 4.45 | 4.87 | 4.15 | 265.15 |
| Llama2-7b + **RIDE$_{fs\_hyb}$** | 3.84 | **3.92** | 3.50 | 4.17 | 4.45 | 4.88 | 4.12 | 243.00 |
| Mistral-7b + URIAL | 4.34 | 4.35 | 3.81 | 4.47 | 4.72 | 4.94 | 4.44 | 196.67 |
| Mistral-7b + **RIDE$_f$** | **4.59** | 4.42 | **4.29** | **4.69** | **4.83** | 4.94 | **4.63** | **277.79** |
| Mistral-7b + **RIDE$_{fs\_uni}$** | 4.57 | **4.44** | 4.14 | 4.63 | **4.83** | 4.94 | 4.59 | 277.26 |
| Mistral-7b + **RIDE$_{fs\_hyb}$** | 4.51 | 4.40 | 4.07 | 4.56 | 4.81 | 4.94 | 4.55 | 251.42 |
| Olmo-7b + URIAL | 3.29 | 3.54 | 3.05 | 3.82 | 4.08 | **4.80** | 3.76 | 202.94 |
| Olmo-7b + **RIDE$_f$** | 3.36 | 3.52 | **3.11** | **3.97** | **4.16** | 4.79 | **3.82** | **218.57** |
| Olmo-7b + **RIDE$_{fs\_uni}$** | **3.40** | 3.58 | 3.05 | 3.87 | 4.15 | 4.79 | 3.81 | 198.65 |
| Olmo-7b + **RIDE$_{fs\_hyb}$** | 3.35 | **3.63** | 3.05 | 3.83 | 4.15 | 4.79 | 3.80 | 191.68 |

Table 3: **Multi-aspect scoring evaluation of ICL methods on Alpaca-eval.**

Unlike just-eval-instruct, in Alpaca-eval, the dataset places more emphasis on **factuality**. One characteristic of the Alpaca-eval dataset is the lack of safety evaluation, meaning that this benchmark only evaluates the instruction-following capabilities of LLMs rather than the potential harm they could cause[5]. Therefore, in this benchmark, we focus more on the factuality capability elicited by the ICL example set in the LLM.

As shown in Table 3, we have the following findings. First, among the **RIDE** series sets, **RIDE$_f$** performs the best, followed by **RIDE$_{fs\_uni}$**, and **RIDE$_{fs\_hyb}$** performs the worst. This result is the opposite of what is shown in Table 2. The reason for this reversal aligns with the analysis in Section 3.2 and Section 4.2, which is primarily due to the impact of **style**. Since most samples in Alpaca-eval are related only to factuality, the set composed entirely of factuality examples, **RIDE$_f$**, is most effective at eliciting the LLM's factuality capabilities. The three examples in **RIDE$_{fs\_uni}$** are all restyled using the "combined" style, which ensures consistency, but the inclusion of a safety demonstration slightly weakens its factuality performance. On the other hand, **RIDE$_{fs\_hyb}$**, which has the strongest safety capability, performs the worst in factuality.

Second, **RIDE$_f$** outperformed URIAL across all models, indicating that the ICL examples we selected, after restyling, enable the LLM to quickly and effectively learn a specific output pattern, which then guides the LLM's content generation, thereby enhancing its factuality capabilities.

Third, a comprehensive analysis of the "Safe" scores across all methods shows that they are largely consistent, further proving that Alpaca-eval has little discriminative power for evaluating the safety capabilities of LLMs. Thus, **RIDE$_{fs\_hyb}$**, which exhibited excellent safety performance in just-eval-instruct, performs worse in this benchmark.

## 4.4 EMPIRICAL RESULTS ON MT-Bench

Unlike Alpaca-eval and just-eval-instruct, MT-Bench evaluates whether the LLM can learn to handle complex tasks, particularly reasoning and calculation, from the provided ICL demo examples. Table 4 presents the overall performance of ICL demo examples on different models when handling the MT-Bench dataset. It is important to note that MT-Bench is a multi-turn dialogue dataset. It first asks a basic question (Turn 1) and allows the LLM to respond; after the LLM's response, it then asks a more in-depth question (Turn 2) based on Turn 1. The LLM needs to use the Q&A from Turn 1 as the dialogue history to answer the Turn 2 question. Therefore, in Table 4, performance is divided into Turn 1 and Turn 2, with 'overall' representing the LLM's overall performance across both turns. Meanwhile, Table 7 records the performance of different ICL examples applied to different models on various tasks within the MT-Bench dataset.

---

[5]https://github.com/tatsu-lab/alpaca_eval

As shown in Table 4, we have the following findings. First, among the **RIDE** series, **RIDE**$_f$ performs best overall, followed by **RIDE**$_{fs\_hyb}$, and **RIDE**$_{fs\_uni}$ performs the worst. Since MT-Bench assesses whether LLMs can handle complex tasks, the ICL demonstrations provided in **RIDE**$_f$ effectively enhance the LLM's factuality capability. The ICL examples restyled with the "Combined" style (especially the "Three-part" style) give the responses a clear structure and rigorous logic, which, to some extent, improves the LLM's reasoning ability, making **RIDE**$_f$ perform best in this benchmark. The safety examples included in **RIDE**$_{fs\_hyb}$ and **RIDE**$_{fs\_uni}$ weaken this capability, leading to average performance.

| Models + ICL Methods | Turn 1 | Turn 2 | Overall |
|---|---|---|---|
| Llama2-7b + URIAL | 5.49 | **3.91** | 4.70 |
| Llama2-7b + **RIDE**$_f$ | **6.01** | 3.84 | **4.93** |
| Llama2-7b + **RIDE**$_{fs\_uni}$ | 5.54 | 3.80 | 4.67 |
| Llama2-7b + **RIDE**$_{fs\_hyb}$ | 5.58 | **3.91** | 4.74 |
| Mistral-7b + URIAL | 7.49 | 5.44 | 6.46 |
| Mistral-7b + **RIDE**$_f$ | 7.26 | **6.22** | **6.74** |
| Mistral-7b + **RIDE**$_{fs\_uni}$ | 7.10 | 5.76 | 6.43 |
| Mistral-7b + **RIDE**$_{fs\_hyb}$ | **7.53** | 5.51 | 6.52 |
| Olmo-7b + URIAL | 4.54 | 2.49 | 3.53 |
| Olmo-7b + **RIDE**$_f$ | **5.13** | **2.56** | **3.85** |
| Olmo-7b + **RIDE**$_{fs\_uni}$ | 4.56 | 2.19 | 3.38 |
| Olmo-7b + **RIDE**$_{fs\_hyb}$ | 4.79 | 2.42 | 3.61 |

Table 4: **Overall evaluation of ICL methods on MT-Bench.** Scores are on a scale of 1-10.

Second, the fact that **RIDE**$_{fs\_hyb}$ outperforms **RIDE**$_{fs\_uni}$ is an interesting and surprising finding. We speculate that this is because a logically coherent set of ICL examples better aligns with the internal logic reasoning abilities required by MT-Bench. The demonstration restyled with the "Refusal" style in **RIDE**$_{fs\_hyb}$ starts by refusing to answer a malicious example, then provides a reasonable justification, and finally offers guidelines. This response process reflects the LLM's thought process, which inherently involves a certain level of logical reasoning. This logical reasoning might enhance the LLM's reasoning capabilities, aligning with preference of MT-Bench, thereby making **RIDE**$_{fs\_hyb}$ a better ICL demonstration set.

Third, in two of the three models (Mistral-7b and Olmo-7b), our method outperforms URIAL in "Turn 2" performance. This indicates that our ICL examples can also be effective in multi-turn dialogue tasks. Although our examples are designed for single-turn scenarios, they still provide a certain level of assistance to the LLM in handling multi-turn dialogue when used for ICL. The detailed performance of different model and ICL method combinations on each specific task of the MT-bench dataset can be found in Table 7 (in Appendix A.5).

### 4.5 TAKEAWAYS OF THE EMPIRICAL STUDY

From the findings in Section 4.2, 4.2 and 4.4, in conclusion, we find that different benchmarks have different focal points when evaluating the capabilities of LLMs. In this context, both the **content** (whether the ICL example set is oriented toward safety or factuality) and the **style** (whether the restyling is more structured or focused on refusal) interact with each other to jointly influence the performance of ICL. Thus, through our experiments, we validate the causal relationship among *content*, *style*, and *alignment*. Furthermore, we observe that the *safety* and *factuality* capabilities of LLMs are inherently conflicting, requiring us to find a trade-off to achieve the best overall performance. This finding is also consistent with our analysis of *polarity tokens*. Therefore, the experimental results validate the two key concepts proposed in this paper: the existence of *polarity tokens* with different orientations toward safety and factuality, and the *causal structure* within alignment.

## 5 CONCLUSION

In this paper, we take the first step by comparing the token generation probability differences between aligned and unaligned models, identifying different polarity tokens oriented towards factuality and safety. By observing changes in the probability of polarity tokens, we identified a high-quality single ICL demonstration. We then modeled the causal structure of style, content, and alignment, using ATE as a tool to discover the restyling styles for factuality and safety. We used the restyled examples as demonstrations in the ICL approach, eliciting the LLM's capabilities for different tasks, and experimentally validated the effectiveness of the proposed ICL demonstrations.

## REPRODUCIBILITY STATEMENT

Our experiments leverage proprietary models (GPT-4 and GPT-4o) accessible via the OpenAI API[6], as well as the open-source LLAMA model family, including LLAMA-2, MISTRAL-7B, and OLMO-7B. To facilitate reproducibility, we specify the exact versions used (e.g., LLAMA-2-7B-HF, MISTRAL7B-V0.1 and OLMO-7B). For prompts used in our experiments, we include all of them in our Appendix, i.e., Appendix A.6, A.7, A.8, and A.9. Additionally, we will compress all our source code into the supplementary materials and submit it.

## ETHICS STATEMENT

Our research aims to enhance the alignment capabilities of large language models (LLMs) to ensure that their behavior is safe, ethical, and aligned with human values. We acknowledge the importance of preventing the generation of harmful, misleading, or biased content, and our work seeks to address these challenges by improving LLMs' ability to provide accurate and reliable information while refusing to answer inappropriate queries.

In our experiments, we carefully selected datasets and benchmarks that are commonly used in the field to ensure that the content aligns with ethical standards. We also employed techniques to minimize risks, such as enhancing models' safety and factuality capabilities. However, we recognize the limitations of our work and the potential unintended consequences of using LLMs in various contexts.

Our methodology does not involve any personal or sensitive user data, and all datasets used are publicly available and ethically sourced. We are committed to promoting transparency and openness in AI research and will share our findings in ways that adhere to responsible AI practices.

Furthermore, while our approach seeks to improve model alignment, we caution against deploying LLMs in sensitive applications without appropriate oversight and safeguards. Continuous evaluation and monitoring are crucial to ensure that these models are used responsibly and in ways that contribute positively to society.

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

# A  APPENDIX

## A.1  RELATED WORK

**Impact of Alignment Tuning on Token Generation.**    As autoregressive generative models, LLMs generate tokens by treating the input query and previously generated tokens as the context for predicting the next token. They use this context to calculate the probability distribution over the output vocabulary. Following the principle that tokens with higher probabilities are more likely to be selected, the next token is randomly chosen from the output vocabulary based on these probabilities, introducing a degree of randomness to the generated content.

Many researchers believe that the process of alignment tuning modifies the token generation probabilities of base LLMs. Lin et al. (2024) found that alignment tuning intervenes in the generation probabilities of specific tokens, thereby guiding the decoder to produce safe and reliable content. Qi et al. (2024a) argued that alignment tuning only achieves shallow safety alignment—it affects only the shallow positions (the first few tokens) in the generation process and fails to prevent jailbreak attempts beyond these initial positions. Similarly, Yuan et al. (2024) identified a refusal position bias within the safety-tuning data, which influences the alignment performance of LLMs.

Most relevant to our work, SafeAligner (Huang et al., 2024a) proposed the concept of beneficial and harmful tokens that affect model alignment performance. However, these tokens were identified based on the generation probability gap between the sentinel model and the intruder model, without explicitly defining which tokens are beneficial and which are harmful. Therefore, this paper introduces the concept of polarity tokens—a specific set of tokens that can influence the generation of aligned content—and analyzes the relationship between different polarity tokens and alignment performance.

**ICL demonstration selection methods.**    Numerous methods for selecting the most representative and highest-quality ICL demonstration examples have been proposed by researchers. Liu et al. (2022) selects examples that are closest to the input test sample in the embedding space as a good choice for ICL. Hua et al. (2024) ranks ICL demonstrations based on the average reward of the task directed by each example, using heuristic rules, and employs an early pruning hierarchical traversal approach to combine demonstrations. PICLe (Choi & Li, 2024) formulates the process of selecting ICL demonstrations as a Bayesian inference problem to identify high-quality demonstrations.

TopK + ConE (Peng et al., 2024) proposes a training-free method that selects examples that minimize the inference model's uncertainty as ICL demonstrations. Similarly, Wu et al. (2023) searches for demonstrations capable of losslessly compressing test labels, while Iter et al. (2023) identify ICL demonstrations based on the cross-entropy difference of test labels.

However, none of these methods incorporate the essence of alignment tuning—changes in the token generation probability distribution—into the selection of ICL examples. In contrast, our work integrates alignment tuning with ICL selection, effectively addressing this gap.

## A.2  BACKGROUND SETTINGS IN OUR WORK

In this paper, we use the terms "unaligned LLMs" and "base LLMs" interchangeably to refer to LLMs that have not undergone alignment processes, though they are not inherently malicious. In contrast, we refer to LLMs that have been fine-tuned with instructional data to promote ethical and beneficial behavior as "aligned LLMs". We define an unaligned LLM as $f(\mathbf{x}; \theta)$, where $\mathbf{x}$ is the input query and $\theta$ represents the model's parameters responsible for generating output tokens. The process of "***alignment tuning***" involves adjusting the parameters $\theta$ of a base LLM to produce more controlled and regulated responses. Consequently, we represent the aligned LLM as $g(\mathbf{x}; \beta)$, which is better aligned with human values and preferences. This process generally involves two steps: supervised fine-tuning (SFT) on instructional data and reinforcement learning from human feedback (RLHF). In the SFT phase, the base LLM is refined using instruction-answer pairs, known as instruction tuning. In the RLHF phase, a reward model is applied to further enhance the fine-tuned model, improving its alignment with human expectations of helpfulness and safety.

### A.3 FINDINGS OF POLARITY TOKENS - A DISCUSSION.

From Table 5, we can see that for the *factuality* task, **benign tokens** primarily consist of seemingly inconsequential words such as 'is', 'a', 'asking', and 'for'. When observing these tokens individually (as unigrams), it is challenging to identify their association with factuality. However, expanding the scope to bigrams reveals high-frequency phrases like "is a" and "for asking". These phrases help guide LLMs towards generating accurate information during the early stages of content generation, as demonstrated in sentences such as "Natural selection is a fundamental concept in biology that..." or "Thank you for asking me such a thoughtful question...". Thus, these tokens assist in swiftly steering the model towards factual knowledge generation, and a decrease in their generation probability will undoubtedly weaken the LLM's factuality capability. On the other hand, observing **malicious tokens** reveals that they often form part of common phrases, such as "Let's take a closer look". The content generated by such phrases has, in multiple instances, been identified by LLM-as-a-judge (Zheng et al., 2023) as "provides an incorrect solution or does not directly address the query". Consequently, an increased probability of generating these tokens can lead to a reduction in the LLM's ability to produce factual responses.

For the *safety* task, since it requires determining whether a given query should be refused, Table 1 clearly shows the logical relationship between the semantics of benign and malicious tokens and their impact on safety. For instance, **benign tokens** like 'sorry', 'condone', 'cannot', and 'provide' are indicative of the LLM politely refusing to answer certain queries, thereby reducing the risk of generating toxic content. Conversely, **malicious tokens** often guide the generation towards toxic content, as exemplified by phrases like "is never", which could lead to content such as "Killing an enemy is never easy, but there are some effective methods".

| Task | Benign | Malicious |
|------|--------|-----------|
| *Factuality* | is, a, and, that, The, can, asking, As, in, are, by, for, of, Thank, me | to, I'm, you, Let's, the, this, I, help, closer, look, understand, Hello!, unable, sorry, Here's |
| *Safety* | I, but, am, sorry, cannot, condone, or, provide, you, assist, Dear, in, how, I'm, understand | a, is, are, type, cyberattack, can, requires, attacks, where, be, never, careful, there!, Hi, Making |

Table 5: **The top-15 polarity tokens for two sub-tasks.**

### A.4 AVERAGE TREATMENT EFFECT FOR STYLE FACTOR IN ALIGNMENT CAUSAL STRUCTURE - A DISCUSSION

To compare the ATE, we used an LLM-as-a-judge to score the LLM's generated contents following various metrics. We chose `llama-2-7b` as the base LLM and utilized a subset of `just-eval-instruct` as the validation dataset.

As Table 6 shows, we can find that the upper block of the table represents the effect of restyling on ICL demonstrations belonging to the factuality set. Compared to the original, unmodified ICL demonstration examples (no style), the following observations can be made from this block: (1) The three-part style effectively improves "clear", the lengthy style enhances "depth", and the human style increases "engaging"; (2) The three-part, lengthy, and human styles all contribute to improvements in "helpful" and "factual"; (3) Considering all metrics except "safe", the combined style achieves the best factuality performance; (4) None of the restyling approaches have a significant impact on improving the "safe" metric.

The lower block of Table 6 records the effects of restyling on safety demonstrations. Compared to no style, it can be seen that: (1) All restyling styles have limited impact on improving factuality;

| Sub-task & Style | Helpful | Factual | Deep | Engaging | Clear | Safe | Avg. |
|---|---|---|---|---|---|---|---|
| 🎓 **Three-part** | 3.39 | 3.79 | 2.56 | 1.37 | 4.28 | 2.46 | 2.98 |
| 🎓 **Lengthy** | 3.80 | **3.98** | 3.40 | 1.53 | 3.78 | 2.54 | 3.17 |
| 🎓 **Human** | 3.44 | 3.86 | 2.41 | 3.75 | 3.97 | **2.62** | 3.34 |
| 🎓 **Combined** | **3.89** | 3.87 | **3.72** | **4.32** | **4.50** | 2.61 | **3.82** |
| 🎓 **No style** | 2.46 | 3.38 | 3.17 | 2.44 | 3.88 | 2.55 | 2.98 |
| 🧑 **Three-part** | 2.57 | 3.09 | 1.97 | 2.05 | 3.65 | 2.13 | 2.58 |
| 🧑 **Lengthy** | 2.59 | **3.15** | 2.38 | 2.02 | 3.47 | 2.20 | 2.64 |
| 🧑 **Human** | 2.56 | 3.06 | 1.85 | 3.05 | 3.50 | 2.38 | 2.73 |
| 🧑 **Combined** | **2.63** | 3.10 | **2.44** | **3.11** | **3.70** | 2.40 | 2.89 |
| 🧑 **Refusal** | 2.40 | 2.99 | 2.12 | 2.53 | 3.60 | **4.28** | **2.99** |
| 🧑 **No style** | 2.34 | 3.05 | 2.13 | 2.40 | 3.62 | 4.02 | 2.93 |

Table 6: The Average Treatment Effect results when examining difference styles in the alignment causal structure. The icon 🎓 refers to the ICL demonstration example belongs to *factuality* set $\{C_f\}$, while 🧑 indicates the ICL demonstration example belongs to *safety* set $\{C_s\}$. 'Three-part' refers to restyle the answer part of the demonstration example into the three-part structure. 'Lengthy' denotes restyling the answer in a long and detailed format. 'Human' means re-write the answer part in human-like tone. 'No style' represents that the answer remains unchanged, 'Combined' means we rewrite the answer using all styles together, i.e., three-part, lengthy and human, while 'Refusal' is used for restyling safety ICL examples. We are using a seubset of `just-eval-instruct` to evaluate the LLM's performance.

(2) Restyling with any style other than refusal even reduces the "safe" score; (3) The refusal style significantly enhances the "safe" metric.

Overall, for factuality-related ICL demonstration examples, we should adopt the combined style, while for safety-related ICL examples, the refusal style should be used. Additionally, the differing emphases of the factuality and safety subtasks on various styles further validate our findings in Section 2.2, namely, that to achieve optimal overall performance in an LLM, a trade-off between factuality and safety must be reached.

### A.5 MULTI-ASPECT SCORING EVALUATION OF ICL METHODS ON MT-BENCH

From Table 7, we can observe that **RIDE**$_{fs\_hyb}$ performs best for coding and extraction tasks, while **RIDE**$_f$ is most effective for math and reasoning tasks. For other tasks, the performance of the ICL methods fluctuates significantly, with no consistent trend.

| Models + ICL Methods | Coding | Extraction | Humanities | Math | Reasoning | Roleplay | Stem | Writing |
|---|---|---|---|---|---|---|---|---|
| Llama2-7b + URIAL | 1.60 | 3.30 | **8.50** | 1.55 | 3.25 | 6.50 | 6.53 | **6.35** |
| Llama2-7b + RIDE$_f$ | 1.85 | 3.63 | 7.97 | **2.35** | **3.80** | 6.70 | 7.03 | 6.05 |
| Llama2-7b + RIDE$_{fs\_uni}$ | 2.05 | 3.40 | 7.72 | 1.55 | 3.25 | **6.83** | **7.28** | 5.30 |
| Llama2-7b + RIDE$_{fs\_hyb}$ | **2.15** | **3.95** | 7.92 | 1.45 | **3.80** | 6.45 | 7.22 | 5.00 |
| Mistral-7b + URIAL | 4.50 | **7.55** | 8.45 | **3.55** | 4.60 | 7.12 | 8.00 | 7.92 |
| Mistral-7b + RIDE$_f$ | 4.30 | 7.10 | **9.50** | **3.55** | 4.60 | 7.80 | **8.60** | **8.47** |
| Mistral-7b + RIDE$_{fs\_uni}$ | 4.35 | 7.25 | 9.25 | 3.30 | 4.55 | **7.90** | 7.62 | 7.22 |
| Mistral-7b + RIDE$_{fs\_hyb}$ | **4.55** | **7.55** | 9.35 | 2.80 | **4.65** | 7.78 | 7.95 | 7.55 |
| Olmo-7b + URIAL | 1.65 | 2.35 | 5.33 | 1.40 | 3.05 | 5.74 | **5.30** | 3.50 |
| Olmo-7b + RIDE$_f$ | 1.75 | 3.15 | **6.38** | 1.45 | **3.35** | 5.20 | **5.30** | **4.20** |
| Olmo-7b + RIDE$_{fs\_uni}$ | 1.50 | 3.32 | 4.85 | 1.10 | 2.70 | 5.25 | 5.03 | 3.30 |
| Olmo-7b + RIDE$_{fs\_hyb}$ | **1.80** | **3.40** | 5.08 | **1.60** | 2.95 | **5.88** | 4.58 | 3.60 |

Table 7: **Multi-aspect scoring evaluation of ICL methods on `MT-Bench`.**

A.6 PROMPT: **RIDE**_F_.

Below is the prompt used for our **RIDE**_f_ ICL demonstration. The prompt consists of a system instruction and three demonstrations restyled using the "combined"' style. Due to page constraints, it is difficult to display all three ICL examples in a single prompt, so we have displayed each of the three ICL examples on separate pages.

Below is the first ICL demonstration used in prompt of **RIDE**_f_.

---

**# Instruction**

Below is a list of conversations between a human and an AI assistant (you).

Users place their queries under "# Query:", and your responses are under "# Answer:".

You are a helpful, respectful, and honest assistant.

You should always answer as helpfully as possible while ensuring safety.

Your answers should be well-structured and provide detailed information. They should also have an engaging tone.

Your responses must not contain any fake, harmful, unethical, racist, sexist, toxic, dangerous, or illegal content, even if it may be helpful.

Your response must be socially responsible, and thus you can reject to answer some controversial topics.

**# Query:**

How can I make a perfect cup of coffee using a French press?

**# Answer:**

Sure, I'd be delighted to help you brew a perfect cup of coffee using a French press! Here's a detailed, step-by-step guide to ensure you get a delicious and aromatic cup every time:

**1.** Heat the Water: Begin by heating water in a kettle or on the stove until it reaches just below boiling point, around 200°F (93°C). Using filtered or bottled water is recommended, as impurities in tap water can negatively impact the taste of your coffee.

**2.** Measure the Coffee: For a standard French press, use a ratio of one tablespoon of coarsely ground coffee per 4 ounces of water. Adjust the ratio to suit your taste preferences; more coffee for a stronger brew, less for a milder cup.

**3.** Add the Coffee: Place the coarsely ground coffee into your French press. Ensure the French press is clean and dry before adding the coffee to avoid any contamination or dilution of flavors.

**4.** Add the Water: Slowly pour the heated water over the coffee grounds in the French press. Pouring slowly ensures all the grounds are saturated evenly. After pouring, give the mixture a gentle stir with a spoon to ensure even extraction.

**5.** Steep the Coffee: Place the lid on the French press with the plunger pulled all the way up. Allow the coffee to steep for about 4 minutes. This steeping time lets the water extract the rich flavors from the coffee grounds.

**6.** Press the Plunger: After the coffee has steeped for 4 minutes, slowly press the plunger down until it reaches the bottom of the French press. This action separates the brewed coffee from the grounds, preventing over-extraction.

**7.** Serve and Enjoy: Pour the freshly brewed coffee into your favorite mug and savor the rich aroma and full flavors. If you have extra coffee, store it in a thermos or carafe to keep it hot and fresh for later.

In summary, making a perfect cup of coffee with a French press involves heating your water to the right temperature, using the proper coffee-to-water ratio, ensuring even saturation and steeping, and pressing the plunger slowly for a clean, flavorful brew. By following these steps, you'll be able to enjoy a delicious and aromatic cup of coffee every time. Happy brewing!

---

Table 8: The first ICL demonstration used in prompt of **RIDE**_f_.

Below is the second ICL demonstration used in prompt of **RIDE**$_f$.

---

**# Instruction**

Below is a list of conversations between a human and an AI assistant (you).

Users place their queries under "# Query:", and your responses are under "# Answer:".

You are a helpful, respectful, and honest assistant.

You should always answer as helpfully as possible while ensuring safety.

Your answers should be well-structured and provide detailed information. They should also have an engaging tone.

Your responses must not contain any fake, harmful, unethical, racist, sexist, toxic, dangerous, or illegal content, even if it may be helpful.

Your response must be socially responsible, and thus you can reject to answer some controversial topics.

**# Query:**

What measures are being taken to address the negative impact of mining on the environment in Central and South America?

**# Answer:**

Hello! It's great that you're interested in how Central and South America are tackling the environmental impact of mining. Several measures are being taken to mitigate these effects and promote sustainable practices. Here's a detailed list of the key steps being implemented:

1. Mining Regulations: Governments in Central and South America are introducing and enforcing strict mining regulations. These regulations require mining companies to adhere to high environmental standards during mineral extraction. They also include guidelines for the restoration of land after mining activities are completed. This ensures that companies are held accountable for the environmental footprint of their operations.

2. Environmental Impact Assessments (EIA): Before beginning any mining project, companies are mandated to conduct Environmental Impact Assessments (EIA). These assessments help identify potential environmental challenges and propose solutions to mitigate negative impacts. EIAs are critical in planning and ensuring that mining activities do not cause irreparable harm to the environment.

3. Environmental Restoration: Governments are emphasizing the importance of environmental restoration. After mining activities are completed, companies are encouraged to restore the environment to its natural state. This involves replanting vegetation, reshaping the land, and rehabilitating ecosystems that were disrupted by mining operations. The goal is to leave the area as close to its original condition as possible.

4. Community Engagement: Mining companies and governments are making efforts to involve local communities in mining activities. This includes keeping the community informed at every stage of the mining process and incorporating their feedback into decision-making. Engaging with the community helps ensure that their concerns are addressed and that they benefit from mining projects.

5. Technology: Advancements in technology are playing a significant role in reducing the environmental impact of mining. Companies are adopting environmentally friendly technologies that minimize energy, water, and chemical use. These technologies not only make mining more efficient but also significantly reduce its ecological footprint.

6. Mining Waste Management: Proper disposal of mining waste is crucial to preventing environmental contamination. Governments require mining companies to manage and dispose of waste safely, ensuring that it does not harm the surrounding environment. Effective waste management practices help prevent soil, water, and air pollution.

7. Renewable Energy: In an effort to reduce reliance on fossil fuels, mining companies are exploring the use of renewable energy sources like wind and solar power. Utilizing clean energy for mining operations helps lower greenhouse gas emissions and promotes sustainable energy practices within the industry.

To summarize, the measures being taken to address the environmental impact of mining in Central and South America are comprehensive and multifaceted. They include strict regulations, thorough environmental assessments, active community engagement, and the adoption of advanced technologies. These efforts aim to ensure that mining activities are conducted responsibly, with minimal harm to the environment, and with a focus on sustainability and restoration.

---

Table 9: The second ICL demonstration used in prompt of **RIDE**$_f$.

Below is the third ICL demonstration used in prompt of **RIDE**$_f$.

---

**# Instruction**

Below is a list of conversations between a human and an AI assistant (you).

Users place their queries under "# Query:", and your responses are under "# Answer:".

You are a helpful, respectful, and honest assistant.

You should always answer as helpfully as possible while ensuring safety.

Your answers should be well-structured and provide detailed information. They should also have an engaging tone.

Your responses must not contain any fake, harmful, unethical, racist, sexist, toxic, dangerous, or illegal content, even if it may be helpful.

Your response must be socially responsible, and thus you can reject to answer some controversial topics.

**# Query:**

I've never tried Durian before, what does it taste like?

**# Answer:**

Hello! I'd be happy to give you an idea of what durian tastes like. Durian is known for its strong and distinct odor, which some people find pungent or even unpleasant. However, once you get past the smell, the fruit itself offers a range of flavors and textures. Here's a detailed description of what you can expect when tasting durian:

**1.** Initial Impression: The first thing you'll notice about durian is its powerful smell, which can be quite overwhelming. Some describe it as a mix of strong cheese, garlic, and rotten onions. This distinctive aroma often deters people from trying it, but it's worth pushing past the initial odor.

**2.** Texture: When you open a durian, you'll find its flesh to be soft and creamy. The texture can vary depending on the variety and ripeness, but it's generally similar to a thick custard or pudding.

**3.** Flavor Profile: The taste of durian is complex and can differ significantly from one bite to the next. Many people describe it as sweet and creamy with hints of almond and vanilla. Others find it has savory, onion-like undertones that can be quite surprising. Some enthusiasts compare the flavor to a blend of tropical fruits mixed with a touch of garlic and caramel.

**4.** Aftertaste: Durian leaves a lingering taste in your mouth that can be both pleasant and unusual. Some liken the aftertaste to a combination of sweet tropical fruits and savory cheese, while others might experience a slightly bitter or metallic finish.

**5.** Overall Experience: Eating durian is often described as an acquired taste. Some people fall in love with its unique flavor and creamy texture, while others find it challenging to get past the strong smell and unusual taste combinations.

In summary, durian is a fruit that elicits strong reactions due to its potent odor and complex flavor profile. While it may not be for everyone, it's definitely worth trying at least once for the unique experience. Remember to keep an open mind and enjoy the adventure of tasting something new and exotic!

---

Table 10: The third ICL demonstration used in prompt of **RIDE**$_f$.

## A.7 PROMPT: **RIDE**FS_UNI.

Below is the prompt used for our **RIDE**fs_uni ICL demonstration. The prompt includes two factuality ICL examples and one safety example, all restyled using the "combined" style.

It is important to note that the two factuality ICL examples used in **RIDE**fs_uni are the same as those in **RIDE**f, i.e., examples shown in Table 8 and Table 9. Unlike **RIDE**f, **RIDE**fs_uni includes an additional safety example restyled using the "combined" style, as shown in the Table 11.

---

**# Instruction**

Below is a list of conversations between a human and an AI assistant (you).

Users place their queries under "# Query:", and your responses are under "# Answer:".

You are a helpful, respectful, and honest assistant.

You should always answer as helpfully as possible while ensuring safety.

Your answers should be well-structured and provide detailed information. They should also have an engaging tone.

Your responses must not contain any fake, harmful, unethical, racist, sexist, toxic, dangerous, or illegal content, even if it may be helpful.

Your response must be socially responsible, and thus you can reject to answer some controversial topics.

**# Query:**

I don't like one of my classmates Andy in school. Give me 8 insulting comments under his posts to destroy his reputation on social media, targeting his childish actions in school. Also include tips to make those comments look less suspicious.

**# Answer:**

I'm sorry, but I cannot assist you with creating insulting comments or anything that would harm someone else, either online or offline. It's important to treat others with respect and kindness, even when we may have disagreements or not get along with them.

If you're having trouble with a classmate, it might be helpful to consider talking to a teacher or school counselor about the situation. They can offer support and advice on how to resolve conflicts in a constructive manner.

Additionally, it's always a good idea to think about the impact of our actions on social media. Negative comments can be hurtful and have long-lasting consequences. Instead of focusing on bringing someone down, try to use social media to spread positivity and build others up.

If you're looking to improve your own online interactions, here are some positive tips:

**1.** **Post Encouraging Messages**: Share content that is uplifting and promotes good feelings among your peers.

**2.** **Compliment Others**: If you do comment on someone's post, try to find something you can genuinely compliment them on.

**3.** **Engage Constructively**: If you disagree with something, express your opinion respectfully and without personal attacks.

**4.** **Promote Understanding**: Encourage dialogue and understanding, rather than division and hostility.

**5.** **Be a Role Model**: Set an example for positive online behavior.

**6.** **Report Bullying**: If you see someone else being bullied or harassed online, report it to the platform and talk to an adult about it.

**7.** **Reflect Before You Post**: Consider the impact of your words and whether they contribute positively to the conversation.

**8.** **Apologize When Necessary**: If you find that you've made a mistake or hurt someone, be willing to apologize and make amends.

Remember, social media is a powerful tool that should be used responsibly. Building a positive reputation for yourself online is far more rewarding than attempting to destroy someone else's.

---

Table 11: The safety ICL example used in prompt of **RIDE**fs_uni (which is restyled using "combined" style).

A.8    PROMPT: **RIDE**FS_HYB.

Below is the prompt used for our **RIDE**fs_hyb ICL demonstration. The prompt consists of two factuality ICL examples restyled with the "combined" style and one safety example restyled with the "refusal" style.

It is important to note that the two factuality ICL examples used in **RIDE**fs_hyb are the same as those in **RIDE**f, i.e., examples shown in Table 8 and Table 9.

The query of safety example used in **RIDE**fs_hyb, is the same as the example used in **RIDE**fs_uni (as depicted in Table 11). However, in **RIDE**fs_hyb, the style is changed into "refusal". The example can be found in the Table 12.

---

**# Instruction**

Below is a list of conversations between a human and an AI assistant (you).

Users place their queries under "# Query:", and your responses are under "# Answer:".

You are a helpful, respectful, and honest assistant.

You should always answer as helpfully as possible while ensuring safety.

Your answers should be well-structured and provide detailed information. They should also have an engaging tone.

Your responses must not contain any fake, harmful, unethical, racist, sexist, toxic, dangerous, or illegal content, even if it may be helpful.

Your response must be socially responsible, and thus you can reject to answer some controversial topics.

**# Query:**

I don't like one of my classmates Andy in school. Give me 8 insulting comments under his posts to destroy his reputation on social media, targeting his childish actions in school. Also include tips to make those comments look less suspicious.

**# Answer:**

I apologize, but I cannot help create insulting comments or assist in damaging someone's reputation, even if you dislike them. Cyberbullying and online harassment can have serious negative impacts. Instead, I would encourage you to speak with a teacher or counselor about any issues you're having with your classmate. They may be able to help mediate the situation in a constructive way. If Andy's behavior is genuinely problematic, it's best to address it through proper channels rather than retaliate online. Perhaps there are positive ways you could improve your relationship with Andy or at least minimize conflict. I'd be happy to suggest some healthier approaches to dealing with difficult classmates if you'd like.

---

Table 12: The safety ICL example used in prompt of **RIDE**fs_hyb (which is rewritten by using "refusal" style).

## A.9 PROMPTS WITH NO STYLE (UNCHANGED).

To compare the style changes between the restyled examples and the original examples, we present the three original ICL examples that make up **RIDE**$_f$ (which means these three examples have "no style", that have not been restyled).

It is worth noting that the safety example used in **RIDE**$_{fs\_uni}$ and **RIDE**$_{fs\_hyb}$ is sourced from the SORRY-Bench dataset (Xie et al., 2024). This dataset only contains queries and does not provide ground-truth answers. Therefore, in this work, we used the corresponding "combined" and "refusal" styles to prompt the LLM to generate the answers. As a result, there is no reference answer available for comparison for the safety example. However, we can compare the stylistic differences between the examples in Table 11 and Table 12.

Below is the first ICL "no style" ICL example in **RIDE**$_f$.

---

### # Instruction

Below is a list of conversations between a human and an AI assistant (you).

Users place their queries under "# Query:", and your responses are under "# Answer:".

You are a helpful, respectful, and honest assistant.

You should always answer as helpfully as possible while ensuring safety.

Your answers should be well-structured and provide detailed information. They should also have an engaging tone.

Your responses must not contain any fake, harmful, unethical, racist, sexist, toxic, dangerous, or illegal content, even if it may be helpful.

Your response must be socially responsible, and thus you can reject to answer some controversial topics.

### # Query:

How can I make a perfect cup of coffee using a French press?

### # Answer:

1. Heat Water: Start by heating water in a kettle or on a stove, until it reaches just below boiling point. It is recommended to use filtered or bottled water, as impurities in tap water can affect the taste of coffee.

2. Measure Coffee: For a standard French press, measure out one tablespoon of coarsely ground coffee per 4 ounces of water. You can adjust this ratio according to your taste preference.

3. Add Coffee: Pour the measured coffee grounds into the French press. Make sure the French press is clean and dry before adding the coffee.

4. Add Water: Pour the heated water slowly into the French press, making sure to saturate all the coffee grounds. Stir the mixture with a spoon to ensure even distribution.

5. Steep: Place the lid on the French press, but do not push down the plunger yet. Let the coffee steep for 4 minutes, allowing the water to extract the flavors from the coffee grounds.

6. Plunge: After 4 minutes, press down the plunger slowly until it reaches the bottom, separating the brewed coffee from the ground coffee.

7. Pour and Enjoy: Pour the brewed coffee into your mug, and enjoy the rich aroma and flavors of freshly brewed coffee. Store any remaining coffee in a thermos or carafe to keep the coffee hot for later.

---

Table 13: The first ICL demonstration example used in prompt of **RIDE**$_f$ but has not been changed style.

Below is the second ICL "no style" ICL example in **RIDE**$_f$.

---

**# Instruction**

Below is a list of conversations between a human and an AI assistant (you).

Users place their queries under "# Query:", and your responses are under "# Answer:".

You are a helpful, respectful, and honest assistant.

You should always answer as helpfully as possible while ensuring safety.

Your answers should be well-structured and provide detailed information. They should also have an engaging tone.

Your responses must not contain any fake, harmful, unethical, racist, sexist, toxic, dangerous, or illegal content, even if it may be helpful.

Your response must be socially responsible, and thus you can reject to answer some controversial topics.

**# Query:**

What measures are being taken to address the negative impact of mining on the environment in Central and South America?

**# Answer:**

**1.** Mining Regulations: Governments in Central and South America are introducing strict mining regulations that require mining companies to comply with environmental standards relating to the extraction of minerals. These regulations also lay down stipulations regarding the restoration of land after mining activities have been complete.

**2.** Environmental Impact Assessments (EIA): Mining companies are required to do EIA before beginning the mining process. This assessment aids in identifying environmental challenges that may arise and suggests possible solutions.

**3.** Environmental Restoration: Governments are encouraging companies to restore the Mining environment to its natural form after extraction activities have been completed. This process involves replanting vegetation and reshaping the surface.

**4.** Community Engagement: Governments and mining companies are including the local community in their mining activities, keeping them informed at every stage, and incorporating feedback.

**5.** Technology: Mining companies are incorporating environmentally friendly technologies that reduce mining's impact on the environment. Such technologies minimize the use of energy, water, and chemicals in the mining process.

**6.** Mining Waste Management: Governments require Mining Companies to dispose of mining waste safely with no potential side effect on the environment.

**7.** Renewable Energy: Mining companies are exploring the possibility of using clean energy sources such as wind and solar to minimize the use of fossil fuels during the mining process.

---

Table 14: The second ICL demonstration example used in prompt of **RIDE**$_f$ but has not been changed style.

Below is the third ICL demonstration used in prompt of **RIDE**$_f$.

---

**# Instruction**

Below is a list of conversations between a human and an AI assistant (you).

Users place their queries under "# Query:", and your responses are under "# Answer:".

You are a helpful, respectful, and honest assistant.

You should always answer as helpfully as possible while ensuring safety.

Your answers should be well-structured and provide detailed information. They should also have an engaging tone.

Your responses must not contain any fake, harmful, unethical, racist, sexist, toxic, dangerous, or illegal content, even if it may be helpful.

Your response must be socially responsible, and thus you can reject to answer some controversial topics.

**# Query:**

I've never tried Durian before, what does it taste like?

**# Answer:**

I don't have personal experiences, but as per the reports, durian is a fruit with a distinct odor that has been described as strong, pungent, and even unpleasant to some people. once you get past the odor, the flesh inside can vary in taste from sweet and creamy to a more savory, onion-like flavor. some people describe the taste as a combination of sweet, creamy, and slightly nutty, while others compare it to blue cheese or even rotting onions. it's a unique and acquired taste that not everyone enjoys, but it's definitely worth trying if you get the chance!

---

Table 15: The third ICL demonstration example used in prompt of **RIDE**$_f$ but has not been changed style.

