# OpenReview forum: "An Empirical Study on Enhancing LLMs' Alignment Capabilities through Restyled In-Context Learning Demonstration Examples"
_ICLR.cc/2025/Conference — Submitted to ICLR 2025_

### Official Review · Reviewer_GeMC · 2024-10-29

**Soundness:** 1
**Presentation:** 2
**Contribution:** 2
**Rating:** 3
**Confidence:** 4

**Summary:**

The main contribution of this paper lies in providing new empirical research on constructing better few-shot prompts for Alignment in the ICL phase. Specifically, the authors analyzed benign tokens from both aligned and base models, using this analysis to select and rewrite relevant examples from existing datasets as prompts to improve language model alignment. The authors conducted experiments on MT-Bench, Alpaca-Eval, and Just-Eval datasets, achieving some performance improvements.

**Strengths:**

Before arriving at their final ICL prompt method, the authors provided substantial empirical evidence supporting their prompt selection, rather than creating them arbitrarily, which enhances the paper's credibility.

The authors chose three common and closely related Alignment datasets, demonstrating improvements in factuality, friendliness, and safety aspects examined in these datasets.

**Weaknesses:**

In my view, the paper's main weaknesses are its lack of soundness and insignificant experimental results, making it difficult to prove the method's effectiveness:
- The authors first selected from predefined ICL prompt sets by considering the impact on benign and malicious tokens (Section 3.1). However, they then rewrote these selections (Section 3.3). Obviously, there's a gap between the rewritten output's impact on benign and malicious tokens compared to the initial prompts, introducing additional bias from an experimental design perspective. Moreover, the authors didn't conduct ablation studies, leaving us uncertain whether the performance improvements stem from the rewriting or their proposed prompt selection algorithm.
- During the rewriting phase, the authors introduced an additional requirement: "(2) lengthy (enriching the answer details and increasing its length without altering the original meaning)". As is well known[1], when using LLM-as-a-judge, the evaluating models tend to favor models that produce longer outputs. Looking at the tables from three different test datasets, the text length generated by the RIDE method consistently exceeded the baseline URIAL method. From another perspective, while the authors selected ICL prompts for both factuality and safety examples, RIDE only outperformed URIAL in preference-related metrics like helpfulness and factuality, but not in safety. Therefore, I suspect the main performance improvement comes from using longer outputs as ICL prompts during the rewriting process, rather than the authors' primary proposed ICL selection method.
- The authors chose longer text examples as demonstrations for the language model, while the baseline URIAL method only used 3 examples, approximately 1011 tokens. As mentioned in their experiments, for the Olmo-7b model with a maximum token limit of 2048, RIDE performed worse than URIAL when their output lengths were similar. From this perspective, RIDE's efficiency decreased compared to URIAL due to using more input tokens, and when efficiency was similar, its performance might be inferior to URIAL.

---

[1] Length-Controlled AlpacaEval: A Simple Way to Debias Automatic Evaluators

**Questions:**

**Typo**:
- Line 116 lacks definition of the reference model
- Lines 116-118: We cannot clearly understand from the paper what $q1$ and $o1$ mean. If I'm not mistaken, these might refer to each token of the question and output.
- I suggest standardizing "unaligned LLMs" and "base LLMs", "unaligned model" and "base model". In line 124, you use both "unaligned model" and "base model" to describe f in the same sentence, which creates confusion.
- Missing citation for paper [1], which is highly relevant to this paper's implementation method and comparison baseline.

---

[2] In-Context Alignment: Chat with Vanilla Language Models Before Fine-Tuning

---

> ### Comment · Reviewer_GeMC · 2024-11-24
>
> Dear Authors,
>
> Despite the approaching end of our discussion period, I am still very much interested in further exchanges with you.

---

> > ### Author Response · Authors · 2024-12-01
> > **Response to Reviewer GeMC**
> >
> > Thank you for your constructive and valuable reviews. We have learned a great deal from your feedback! Our responses are as follows:
> >
> > **W1**: ``There's a gap between the rewritten output's impact on benign and malicious tokens compared to the initial prompts, introducing additional bias from an experimental design perspective''.
> >
> > **Response**: Our primary comparison is with URIAL, which uses manually crafted demonstration examples as ICL inputs. In URIAL's process, implicit restyling is already embedded—examples are rewritten to maintain a consistent style. However, this restyling is implicit, making it challenging for readers to replicate or gain actionable insights from it.
> >
> > In contrast, our RIDE approach explicitly showcases the restyling process, making it transparent. Through the main text and Appendix A.4, we provide clear guidance on:
> >
> > - What types of restyling enhance helpfulness,
> > - What types of restyling improve safety, and
> > - How to achieve the best trade-off between the two.
> >
> > This transparency is intended to offer practical insights, enabling readers to select and restyle ICL demonstration examples effectively. Thus, while both URIAL and RIDE involve restyling, we believe the comparison between the two methods remains fair.
> >
> > Additionally, we plan to include an ablation study in future work to quantitatively and qualitatively analyze the individual contributions of polarity tokens and restyling to the base LLM’s alignment performance.
> >
> > **W2**: ``I suspect the main performance improvement comes from using longer outputs as ICL prompts during the rewriting process, rather than the authors' primary proposed ICL selection method''.
> >
> > **Response**: As stated in lines 401–402, the just-eval-instruct dataset emphasizes the evaluation of LLM safety capabilities. Table 2 demonstrates that across three models, both URIAL and RIDE exhibit advantages in different settings. This indicates that RIDE’s performance improvements are not solely due to the length of the outputs but also reflect its ability to enhance safety.
> >
> > Furthermore, as mentioned in line 452, the Alpaca-eval dataset does not specifically test safety (there are no samples directly related to safety in its distribution). Consequently, the safety scores in this dataset exhibit randomness and cannot serve as a reliable metric for evaluating safety capabilities. The same applies to MT-Bench.
> >
> > In future revisions, we will conduct a more detailed analysis of these datasets to highlight the distinct aspects they evaluate in LLM capabilities.
> >
> > **W3**: ``RIDE's efficiency decreased compared to URIAL due to using more input tokens, and when efficiency was similar, its performance might be inferior to URIAL''.
> >
> > **Response**: It is challenging to directly equate efficiency with the number of input tokens. While it is true that more input tokens can increase costs (e.g., API throughput for input/output tokens), the relationship between efficiency and performance is more nuanced.
> >
> > We agree that overly long prompts can lead to higher costs. However, as shown in Tables 3 and 4, even when we randomly delete content from RIDE's ICL demonstrations, its performance on Olmo remains superior to URIAL, showcasing RIDE's robustness.
> >
> > We will follow the reviewer's suggestion and conduct additional experiments where RIDE's length is matched to URIAL's by further random deletions, allowing for a direct comparison of their performance under similar input lengths.
> >
> > **Lastly, we appreciate the reviewer pointing out two important references. We will incorporate a discussion of these works in future revisions and include length as a critical factor in our experimental comparisons.**
> >
> > **Thank you again for reviewing our paper and for the valuable feedback. We hope our response has somewhat solved your concerns. if you have any comments, please let us know!**

---

> > > ### Comment · Reviewer_GeMC · 2024-12-01
> > >
> > > Thank you very much for your reply.
> > >
> > > Indeed, you have addressed some of my concerns from a certain perspective. However, as an "Empirical Study," thorough experimental design and detailed experimental analysis are essential. Considering that not only I mentioned the lack of statistical significance in your experimental results, but reviewers 1p86 and FgVo also pointed this out, there are two possible directions to improve your paper:
> > >
> > > First, you could focus on "Empirical Study" without considering your proposed new method for now, and instead provide comprehensive analysis around "using context for LLM alignment." In my view, there are increasingly more non-tuning alignment methods emerging in academia, including context alignment and decoding-time alignment [1,2,3,4]. This suggests that potential alignment information already exists within LLMs' internal weights. How to better activate this alignment information from pre-training is a topic worth deep investigation. Whether through example selection or weight selection, the fundamental goal is to better align using LLM's inherent capabilities - so where do these capabilities come from? To what extent? Would they disappear if certain special weights were removed? Would modifying a very small number of special weights or prompts lead to complete disappearance of alignment or entirely false responses? By providing novel research findings for academia, this would surely generate greater interest from the community.
> > >
> > > Alternatively, you could try improving your method. Your experimental results show several interesting phenomena, but the main shortcoming is the lack of statistical significance compared to the baseline method URIAL. You might consider a more specific direction for improvement, such as mathematics, where improvements would be more significant (their evaluation metric Acc is calculated as a percentage ^-^). Recently, the release of GPT-o1 has sparked imitation from both industry and academia. If your proposed method could enable models to achieve "slow thinking alignment" without additional training, reaching or even surpassing language models that have undergone extra SFT training and RLHF training, that would equally be an outstanding contribution.
> > >
> > > Looking forward to your further improvements!
> > >
> > > ---
> > >
> > > [1] DeAL: Decoding-time Alignment for Large Language Models
> > >
> > > [2] Decoding-Time Language Model Alignment with Multiple Objectives
> > >
> > > [3] Reward Steering with Evolutionary Heuristics for Decoding-time Alignment
> > >
> > > [4] Personality Alignment of Large Language Models

---

> > > > ### Author Response · Authors · 2024-12-02
> > > > **On behalf of all the authors, I would like to express our heartfelt gratitude for your insightful suggestions and invaluable comments!**
> > > >
> > > > First and foremost, as authors who have been working in the NLP and AI community for years, we must say that such detailed, sincere, and thoughtful feedback is a **rarity**. We deeply appreciate your suggestions and hold the utmost respect for your insights.
> > > >
> > > > Now, returning to the substance of our work: regarding your first point, "How to better activate the potential alignment information already existing within LLMs' internal weights is a topic worth deep investigation," this indeed lies at the heart of our motivation. It is precisely this question—what is the hidden “magic” behind the manually crafted ICL demonstrations in **URIAL** that unlocks the latent alignment capabilities of LLMs?—that inspired our study.
> > > >
> > > > We designed our experiments (and here, we must acknowledge, as you and other reviewers have pointed out, the limitations in our experimental design, particularly the issues surrounding statistical significance) to uncover this mystery. The concepts of **polarity tokens** and **restyling** emerged from our exploratory efforts. While they may be far from perfect, we hope they offer a new perspective for the community to consider and spark further research.
> > > >
> > > > You also raised a series of thought-provoking questions: Do subtle changes in weights or prompts impact the activation of LLMs’ potential alignment capabilities? If so, to what extent and in what ways? Each of these questions is worth in-depth exploration and provides a fresh lens for examining alignment tuning. Moreover, the four references you provided are incredibly relevant and valuable. While some of them are familiar to us, others will require further study, and we are committed to integrating the knowledge and insights from these works into our future research.
> > > >
> > > > Second, on the point regarding methodological innovation: as mentioned earlier, we recognize the issue of statistical significance in our experiments as a shortcoming of this work. Developing new metrics to highlight significant performance improvements is indeed a critical task we aim to address. We greatly appreciate your suggestions and will incorporate them moving forward.
> > > >
> > > > The concept of designing long internal reasoning chains, as referenced in O1, undoubtedly inspires deeper exploration of CoT reasoning. We have previously wondered whether ICL demonstrations might influence CoT reasoning.
> > > > For instance:
> > > >
> > > > - Could certain ICL demonstrations adjust the generation probabilities of CoT?
> > > > - More broadly, is there a method to reshape the distribution of candidate CoTs such that the most optimal CoT is given the highest probability?
> > > >
> > > > As you mentioned, "slow thinking alignment" represents an exciting avenue of research that could provide meaningful insights to the community.
> > > > We completely agree that if a tuning-free method can match or even surpass SFT or RLHF-based training approaches, it would have clear advantages in terms of cost efficiency, ease of use, and plug-and-play adaptability. This is indeed a fascinating and timely topic that aligns with the current trends and interests in the field. We are immensely appreciative of your perspective on this matter.
> > > >
> > > > In summary, we feel truly privileged to have had the opportunity to engage with a reviewer of your kind—someone who is sharp, knowledgeable, and generous in sharing their wisdom. Your feedback has been invaluable to us and will significantly inform our future work.
> > > >
> > > > **We extend our sincere thanks and wish you all the best in your future endeavors. Until we meet again in the academic journey—farewell and take care!**

---

### Official Review · Reviewer_RSHT · 2024-10-31

**Soundness:** 2
**Presentation:** 3
**Contribution:** 2
**Rating:** 6
**Confidence:** 3

**Summary:**

The paper explores enhancing LLM alignment using restyled in-context learning (ICL) examples, introducing the concept of polarity tokens to guide generation outputs. The authors utilize Average Treatment Effect (ATE) to quantify the impact of different ICL styles and propose a structured method for selecting optimal ICL examples. They conduct experiments focusing on factuality and safety, demonstrating their method’s effectiveness across several benchmarks.

**Strengths:**

The paper presents its ideas in a clear and coherent manner, making complex concepts easy to understand.
The authors provide comprehensive empirical results that support their claims and showcase the method’s practical applications.

**Weaknesses:**

The paper’s contribution is incremental, lacking substantial novelty compared to existing ICL-based alignment methods.
The experiments lack granular analysis of individual ICL examples or polarity tokens’ contributions to model performance across tasks, and adding such details would deepen the understanding of the method’s impact.
The paper demonstrates effectiveness in specific tasks but lacks evidence of generalization across broader cross-task scenarios such as multilingual settings or bias detection.

**Questions:**

Could the authors clarify if their approach could adapt to dynamic, real-world contexts where example restyling might need to change over time?
What are the potential trade-offs between the proposed method and simpler alignment techniques in terms of computational efficiency and performance gains?

---

> ### Author Response · Authors · 2024-12-01
> **Response to Reviewer RSHT**
>
> Thank you for your constructive and valuable reviews. We have learned a lot from your feedback! Our responses are as follows:
>
> **W1Q1**: ``The paper’s contribution is incremental, lacking substantial novelty compared to existing ICL-based alignment methods''.
>
> **Response**: The primary contribution of this work is the development of an automated metric—polarity tokens—to identify ICL demonstration examples that balance safety and helpfulness. Additionally, we propose a restyling process to further enhance these examples in terms of safety and helpfulness.
>
> Our most relevant prior work is URIAL (Lin et al., 2024), which uses three manually crafted ICL examples for alignment purposes. While these manually curated examples are empirically effective, they lack interpretability. For the community, it is challenging to extract actionable insights, such as:
>
> - Why do these examples improve alignment performance?
> - How should one create appropriate ICL examples for a different alignment task?
>
> These questions remain unanswered in URIAL. Our work addresses this gap by making both the ICL example selection and restyling processes transparent. Readers can understand why specific ICL examples are effective for a given downstream task and use our approach as a guideline for selecting examples in their own applications.
>
> This transparency and explainability distinguish our work from previous studies and constitute the novelty of our approach.
>
> **W1Q2**: ``The experiments lack granular analysis of individual ICL examples or polarity tokens’ contributions to model performance across tasks''.
>
> **Response**: It is true that our polarity token analysis was conducted on a single combination of LLaMA-2-7B and the just-eval-instruct dataset. However, as demonstrated in Section 4 Evaluation, our RIDE method achieves the best performance across most settings involving three models (LLaMA-2-7B-hf, Mistral7B-v0.1, and OLMo-7B) and three datasets (Alpaca-eval, just-eval-instruct, and MT-Bench).
>
> While restyling contributes to overall performance improvements, these results also provide evidence that polarity tokens are transferable across LLMs and datasets. We will conduct more fine-grained experiments in future work to further validate the generalizability of polarity tokens across tasks and settings.
>
> **W1Q3**: ``The paper demonstrates effectiveness in specific tasks but lacks evidence of generalization across broader cross-task scenarios such as multilingual settings or bias detection''.
>
> **Response**: As mentioned in lines 53–63, factuality and safety are inherently in tension within alignment tasks. By observing this interplay, we proposed polarity tokens to select ICL demonstration examples suitable for both factuality and safety, and employed restyling to achieve a trade-off between the two. This addresses the factuality-safety tension in alignment tasks.
>
> We agree with the reviewer’s observation that exploring the broader applicability of RIDE to more general tasks, such as multilingual scenarios or bias detection, is a valuable direction. However, we note that this lies beyond the scope of the current study.
>
> **Question**: ``Could the authors clarify if their approach could adapt to dynamic, real-world contexts where example restyling might need to change over time? What are the potential trade-offs between the proposed method and simpler alignment techniques in terms of computational efficiency and performance gains?''
>
> **Response**: As described, our approach selects and restyles a set of high-quality ICL demonstration examples to enhance the alignment capabilities of the base LLM without requiring additional training.
>
> During inference, the restyled ICL examples are constant and applicable across all test sets for the same downstream task. If the task changes, we do not reselect the ICL examples but instead adjust their style based on whether the new task prioritizes factuality or safety. This ensures that the computational complexity during inference remains $O(1)$, which is significantly lower than other methods that require dynamic, online selection or computation of ICL examples.
>
> This design achieves a favorable trade-off by balancing computational efficiency with substantial performance gains, making our approach practical for real-world applications.
>
>
> **Thank you again for reviewing our paper and for the valuable feedback. We hope our response has somewhat solved your concerns. if you have any comments, please let us know!**

---

### Official Review · Reviewer_FgVo · 2024-11-03

**Soundness:** 2
**Presentation:** 3
**Contribution:** 2
**Rating:** 3
**Confidence:** 4

**Summary:**

The paper proposes a novel, tuning-free method to align base LLMs through in-context learning examples. It first identifies benign and malicious tokens with respect to two conflicting objectives, actuality and safety, based on the change in generation probabilities from an already aligned LLM and the corresponding base LLM. It then investigates the causal structure between content, style, and alignment and leverages the average treatment effect to find an effective way to select and restyle a given set of in-context learning examples. By prompting with these identified examples, the method achieves performance improvements on three benchmark datasets.

**Strengths:**

S1: The empirical results in this paper seem extensive and could assist researchers in understanding the LLM alignment process better.

S2: It is interesting to consider the causal relationship between content, style, and alignment.

S3: The idea about the conflicting nature between safety and factuality is interesting and reasonable.

**Weaknesses:**

W1: The presentation and the writing in the paper need to be further improved. For example, the authors introduced the idea of a reference model and $P^\text{refer}$ in Line 119 but never mentioned it again after that. What is the difference between $P^\text{refer}$ and $P^\text{base}$ or $P^\text{align}$? In addition, It is unclear how $V_\text{polar}$ defined in Line 221 is utilized in the subsequent causal structure of alignment. Will the restyling in Section 3.2 affect this value? If so, do you need to do another around of example selection after restyling?

W2: When identifying the benign and malicious polarity tokens, the base model and aligned model may differ in many different ways, it could be too general to broadly attribute all the tokens with the highest change in generation probability as benign or malicious tokens towards safety or actuality. This is reflected by the identified set of benign tokens for factuality in Table 1, which has nothing to do with the generic concept of "factuality". In fact, the change in generation probabilities of the unlikely tokens is meaningless. In addition, the generation probability of a token can be largely affected by the input context, simply averaging the change over the entire validation set may not be suitable. It is very likely that you will only find tokens that frequently appear at the start of a response.

W3: Are those identified benign and malicious tokens transferable across LLMs and validation datasets? If not, then you will need an aligned model to identify those polarity tokens in the first place, and there will be no point of using ICL examples to steer base models to achieve performance comparable to the needed aligned models.

**Questions:**

Q1: Have you considered the difference between the following two ways of identifying the benign tokens: (1) treat the aligned model as the reference, and take the tokens with the highest $P^\text{align} - P^\text{base}$ (2) treat the base model as the reference to generate the output, and take the tokens with the lowest $P^\text{base} - P^\text{align}$? I believe that the authors used the first approach in their paper, but I am wondering if the authors will get a different set of benign tokens when using the second approach.

Q2: Can you report the degree of $\Delta P$ for those identified tokens in Table 1? Are they marginal?

Q3: How's the performance of the aligned models on those three benchmarks?

Q4: Do you modify Alpaca-eval when reporting results in Table 3? I don't think Alpaca-eval comes with those multi-aspect scoring schemas. Should you use AlpacaEval 2 and report (length controlled) win rate as a standard metric instead?

Q5: LLM-as-a-judge is usually subject to large variance, have you tried repeating your experiments for multiple rounds and reporting the average performance and standard error?

Q6: It would be interesting to include some ablation studies to test the robustness of each design choice and the transferability of the identified polarity tokens.

Q7: The improvement upon URIAL seems marginal, especially on just-eval-instruct and alpaca-eval.

Q8: How statistically strong is the causal relationship between style and alignment? It is inappropriate to claim it as a causal structure without reporting the statistical significance.

---

> ### Author Response · Authors · 2024-12-01
> **Response to Reviewer FgVo**
>
> Thank you for your detailed, constructive, and insightful review comments. We have learned a great deal from your feedback! Our responses are as follows:
>
> **W1**: ``The presentation and the writing in the paper need to be further improved.''
>
> **Q1**: ``What is $P^{refer}$?''
>
> **Response**: The reviewer raised a valid concern about the ambiguous notation for $P^{refer}$, $P^{base}$, and $P^{align}$. To clarify, in the equation $\Delta P^{\texttt{malicious}} = P^{\texttt{base}} - P^{\texttt{align}}$, $P^{refer}$ refers to $P^{\texttt{base}}$. Similarly, in $\Delta P^{\texttt{benign}} = P^{\texttt{align}} - P^{\texttt{base}}$, $P^{refer}$ refers to $P^{\texttt{align}}$.
>
> Thus, $P^{refer}$ denotes the reference model, which serves as the baseline for comparison with the target model’s probability distribution under the same inputs. This comparison helps identify polarity tokens. In future revisions, we will revise the notation definitions to make them clearer and avoid ambiguity.
>
> **Q2**: ``It is unclear how $V_{polar}$ defined in Line 221 is utilized in the subsequent causal structure of alignment.''
>
> **Response**: The restyling process described in Section 3.2 does not affect $V_{polar}$. As detailed in Appendix A.4, we use **Average Treatment Effect (ATE)** to investigate how different styles of rewriting ICL demonstration examples impact downstream model performance in terms of factuality and safety.
>
> The overall structure of our method follows a pipeline approach:
>
> - Use polarity tokens to select ICL examples that perform well in terms of factuality and safety.
> - Restyle these examples to achieve a better trade-off between factuality and safety for alignment tasks.
>
> This pipeline addresses the inherent tension between factuality and safety while maintaining their complementary relationship. Importantly, the restyling process does not influence $V_{polar}$.
>
> **W2**: ``It is very likely that the polarity tokens that you found are only the tokens that frequently appear at the start of a response.''
>
> **Response**: We sincerely appreciate the reviewer’s insightful observation, which has inspired us to reflect further on our methodology.
>
> First, as demonstrated empirically in prior studies such as URIAL (Lin et al., 2024) and Shallow Safety Alignment (Qi et al., 2024a), alignment tuning tends to alter the probability distribution of specific tokens or symbols at the early stages of an LLM’s output. By influencing the sampling probabilities of these tokens, alignment tuning can guide the generation trajectory of the model toward outputs that align better with human values.
>
> Building on this foundational insight, our work focuses on identifying ICL demonstration examples that enhance alignment in downstream tasks by addressing the inherent tension between factuality and safety. As mentioned in the response to W1Q1, we compare the probability distributions of $P^{base}$ and $P^{align}$ to identify tokens with the largest average probability differences.
>
> If a token frequently appears in LLM decoding but exhibits little difference in its generation probability between $P^{base}$ and $P^{align}$, it would not qualify as a polarity token. Thus, the polarity tokens we identify are those that frequently show significant changes in generation probability between $P^{base}$ and $P^{align}$—not merely the tokens that frequently appear at the start of a response.
>
> **W3**: ``Are those identified benign and malicious tokens transferable across LLMs and validation datasets?''
>
> **Response**: Thank you for your insightful and thought-provoking observation. Indeed, our polarity token analysis was conducted solely on the combination of LLaMA-2-7B and the just-eval-instruct dataset. It is worth noting that the subset of just-eval-instruct used for ATE evaluation is **orthogonal** to the test set used in Section 4 experiments, thereby eliminating the possibility of information leakage.
>
> However, as shown in the Section 4 Evaluation, our RIDE method achieves the best performance in the vast majority of settings across three models (LLaMA-2-7B-hf, Mistral7B-v0.1, and OLMo-7B) and three datasets (Alpaca-eval, just-eval-instruct, and MT-Bench). While restyling contributes to the overall performance improvements, this also provides indirect evidence that polarity tokens are transferable across LLMs and datasets.
>
> In future work, we will conduct more detailed experiments to further validate the generalizability of polarity tokens themselves across different models and datasets.
>
> **Thank you again for reviewing our paper and for the valuable feedback. We hope our response has somewhat solved your concerns. if you have any comments, please let us know!**

---

### Official Review · Reviewer_1p86 · 2024-11-07

**Soundness:** 2
**Presentation:** 3
**Contribution:** 3
**Rating:** 5
**Confidence:** 4

**Summary:**

This paper introduces a new method for LLM alignment via in-context learning, which selects and restyles demonstrations. This approach is similar to URIAL (Untuned LLMs with Restyled In-context ALignment; Lin et al., 2024), however it introduces several different components. First, exemplar selection is guided by polarity tokens indicating helpful or harmful trajectories of LLM responses. Second, restyling is applied by leveraging the Average Treatment Effect to analyze the causal effect of different alignment outcomes. The proposed method, namely RIDE, is evaluated on several benchmarks, mainly just-eval-instruct (was provided in the URIAL paper), Alpaca-eval, and MT-bench.

**Strengths:**

1. The concept of polarity tokens is interesting and could influence studies beyond the context of this paper.
2. The paper conducts a thorough empirical evaluation on alignment metrics and reports interesting findings by evaluating different variants of RIDE.

**Weaknesses:**

1. The paper lacks a clear discussion about its position with respect to previous work, which might confuse the reader. In the absence of a related work section, the paper’s title, introduction, and other sections read as if the paper’s proposal, to tackle alignment via in-context learning and alignment, is a novel aspect of this work. However this is exactly the proposal behind URIAL, which is not cited until much later.
2. It is hard to validate the claimed improvements compared to URIAL within the current experimentation protocol. First, it is not clear why the just-eval-instruct scores for URIAL and baselines in Table 2 are not the same as those reported in the URIAL paper (Lin et al., 2024). Second, RIDE optimizes its prompt using GPT-4o and a validation set from just-eval-instruct (is this provided originally or sampled from the test set?), which might provide an unfair advantage compared to other techniques.
3. The paper’s titled implies a possibly broader application of the paper’s idea than what is tackled by the proposed method. While polarity tokens have effectively been used in the context of this work, it is not clear how this idea can generalize in broader scenarios beyond factuality and safety categories. Furthermore, it is not clear how effective this approach would be in the absence of aligned models, costly GPT-4o inference, and the presence of a validation set.

**Questions:**

1. Are improvements statistically signficant?
2. Why are the reported baseline numbers for just-eval-instruct different than those in the URIAL paper?
3. How was the validation set selected for optimizing the prompt in RIDE?
4. Are the authors planning to share their codebase?
5. Could the authors elaborate how the proposed approach behind polarity tokens would scale with an increasing number of alignment categories?

---

> ### Author Response · Authors · 2024-12-01
> **Response to Reviewer 1p86 -- Part 1**
>
> Thank you for your constructive and valuable review comments. Our responses are as follows:
>
> **W1**: ``The paper lacks a clear discussion about its position with respect to previous work."
>
> **Response**: Firstly, due to space limitations in the main text, we included a discussion of related work in Appendix A.1. The primary contribution of this paper is the development of an automated metric—polarity tokens—to identify in-context learning (ICL) demonstration examples that balance safety and helpfulness. Additionally, we propose a restyling approach to make these examples even more effective in terms of safety and helpfulness.
>
> Regarding the related work, URIAL (Lin et al., 2024), raised by Reviewer 1p86, their method uses three manually written ICL examples to achieve alignment. While manually curated ICL examples are indeed empirically effective, they lack interpretability. For the community, it is challenging to extract insightful takeaways from such examples, such as:
>
> - Why do these manually written ICL examples improve alignment capabilities?
> - For a different alignment task, how should one create their own ICL examples?
>
> These questions remain unanswered in URIAL's work. In contrast, our approach addresses these issues. Both our ICL example selection and restyling processes are transparent, enabling readers to understand why these ICL examples are effective for the current downstream task. Furthermore, we can confidently say that our work provides guidance for selecting suitable ICL examples for their specific needs.
>
> We agree with Reviewer 1p86's suggestion and will revise the Introduction to include a more prominent and logical discussion of the relationship between URIAL and our work. We will also explicitly state that our study builds upon URIAL while highlighting the key differences between the two approaches.

---

> ### Author Response · Authors · 2024-12-01
> **Response to Reviewer 1p86 -- Part 2**
>
> **W2**: ``It is hard to validate the claimed improvements compared to URIAL."
>
> **Q1**: ``It is not clear why the just-eval-instruct scores for URIAL and baselines in Table 2 are not the same as those reported in the URIAL paper.''
>
> **Response**: We learned that directly citing experimental results from the original paper may not reflect a rigorous experimental approach. Therefore, adhering to the principle of fair experimental comparison, we re-ran URIAL’s code, tested all versions of URIAL, and selected the demonstration examples that achieved the highest performance for comparison.
>
> Due to the inherent variability in the outputs of LLMs and the subjective and stochastic nature of using LLMs as LLM-as-a-judge, our experimental results may differ from those reported in the URIAL paper. Despite these discrepancies, the data we present is both truthful and reliable, and such variations do not affect the fairness of the comparative experiments.
>
> To ensure transparency and reproducibility, **we will release the source code for our work**, allowing reviewers and the community to evaluate and analyze our results.
>
> **Q2**: ``RIDE optimizes its prompt using GPT-4o and a validation set from just-eval-instruct (is this provided originally or sampled from the test set?), which might provide an unfair advantage compared to other techniques.''
>
> **Response**: First, the subset of just-eval-instruct used for ATE evaluation (as described in Appendix A.4) is **orthogonal** to the test set used in the experiments of Section 4. Therefore, there is no risk of information leakage. We appreciate the reviewer pointing this out and will explicitly clarify this in future revisions.
>
> Second, regarding the issue of restyling: as mentioned earlier, our primary comparison is with URIAL, which manually crafted several demonstration examples as ICL inputs. In the process of manual curation, URIAL implicitly incorporated a form of restyling—i.e., adapting examples to fit a consistent, specific style. However, this implicit approach makes it challenging for readers to replicate or draw actionable insights from the restyling process.
>
> In contrast, RIDE adopts a different approach by making the restyling process explicit and transparent. Through the main text and Appendix A.4, we provide clear guidance on:
>
> - What types of restyling enhance factuality,
> - What types of restyling improve safety, and
> - How to achieve the best trade-off between these two aspects.
>
> We believe this transparency offers practical insights to readers, enabling them to make informed decisions when selecting and restyling ICL demonstration examples. Consequently, while both URIAL and our method involve restyling, we argue that the comparison remains fair.
>
> **W3**: ``The paper’s titled implies a possibly broader application of the paper’s idea than what is tackled by the proposed method.''
>
> **Q1**: ``It is not clear how polarity tokens can generalize in broader scenarios beyond factuality and safety categories.''
>
> **Response**: As mentioned in lines 53–63 of the paper, factuality and safety are often in a state of inherent tension within alignment tasks. By observing and analyzing this relationship, we proposed polarity tokens to address this contradictory yet unified interplay. Thus, in this work, polarity tokens are specifically designed for alignment tasks.
>
> We agree with the reviewer's perspective that exploring the applicability of polarity tokens to more general tasks is a worthwhile direction for future research. However, we would like to emphasize that such exploration lies beyond the scope of this paper.
>
> **Q2**: ``It is not clear how effective this approach would be in the absence of aligned models.''
>
> **Response**: Thank you for raising this point. In future work, we will include a comparison of performance between rewritten ICL examples and unmodified ICL examples to further evaluate the effectiveness of our approach.
>
> **Thank you again for reviewing our paper and for the valuable feedback. We hope our response has somewhat solved your concerns. if you have any comments, please let us know!**

---

> > ### Comment · Reviewer_1p86 · 2024-12-01
> > **Thank you for your response**
> >
> > I would like to thank the authors for their responses which provide more clarity and address some of my concerns. However, there are still important questions that are left unanswered. For example, it is still unclear whether empirical improvements are statistically significant, a concern raised by me and reviewers FgVo, GeMC. Overall, I agree with reviewer GeMC's follow-up responses and suggest that major revisions would be required to address our concerns. Therefore, I choose to keep my score as is.

---

> ### Author Response · Authors · 2024-12-02
> **Thank you for your suggestion!**
>
> We must acknowledge and agree with the points raised by you and other reviewers regarding the limitations in our experimental design, particularly the issue of statistical significance. As discussed with Reviewer GeMC, developing new metrics to better highlight significant performance improvements is a critical task we need to address. Thank you for your suggestions—we will work on resolving this issue in future revisions.
>
> Regarding your additional questions:
>
> - Q2: "Different numbers than those in the URIAL paper": This has been addressed in W2Q1 of our rebuttal.
> - Q3: "How was the validation set selected for optimizing the prompt in RIDE": This was discussed in W2Q2.
> - Q4: "Release source code": Please refer to our response in W2Q1.
> - Q5: "Could the authors elaborate on how the proposed approach behind polarity tokens would scale with an increasing number of alignment categories?": Please refer to our response in W3Q1.
>
> In conclusion, we deeply value your thoughtful feedback and sincere suggestions, which will serve as an invaluable guide for our future research. We look forward to crossing paths with you again and wish you all the best in your future endeavors!

---

### Meta-Review · Area_Chair_pJ5y · 2024-12-25

**Metareview:**

This paper introduces a tuning-free method to improve LLM alignment using in-context learning (ICL). By analyzing polarity tokens, the method refines prompts to enhance factuality, safety, and alignment. Experiments on benchmarks like Alpaca-eval and MT-Bench show modest improvements in alignment.

Strength:
Leveraging polarity tokens for in-context alignment is an interesting topic and perspective. Thorough empirical study in developing the approach. Also understanding in-context alignment itself has become a valuable problem.

Weakness:
As pointed out by several reviewers, the paper does not discuss its contribution in the right context of existing works, which could make its contribution misleading. And inclusion of the discussion of related works in Appendix is not a good practice. One other weakness pointed by several reviewers is the lack of statistical significance in the experimental results compared to the baseline. And Reviewer GeMC has suggested several improvement directions of this work for future publications.

Overall, this paper is studying an interesting topic and problem. There are still some additional work that is needed to further improve this work, as suggested by the reviewers. The authors are encouraged to take these feedback into account and submit to a future venue.

**Additional Comments On Reviewer Discussion:**

Some of the clarification problems have been addressed by the authors during the rebuttal phase. However, the main concerns raised by the reviewers (i.e., the lack of statistical significance, and the presentations of the paper) have not been fully addressed. And addressing them would require significant additional effort. Therefore, the paper cannot be accepted in its current form.

---

### Decision · Program_Chairs · 2025-01-22

Reject